# GapONet: Nonlinear Operator Learning for Bridging the Humanoid Sim-to-Real Gap

## Abstract

The sim-to-real gap, arising from imperfect actuator modeling, contact dynamics, and environmental uncertainty, poses fundamental challenges for deploying simulated policies on physical robots. In humanoids, object manipulation further amplifies this gap: end-effector payloads alter joint inertia, gravity torques, and transmission efficiency, introducing state- and payload-dependent nonlinearities. Yet existing approaches lack both systematic analysis and a generalizable representation of this payload-induced degradation. To address this limitation, we propose **GapONet**, a payload-conditioned nonlinear operator that maps simulation context functions to residual actions for hardware. We then introduce a payload-aware ⟨collect–analyze–solve⟩ framework to learn this operator **GapONet**. First, we curate a sim-real paired dataset **TWINS** spanning multiple payloads, robots, motions, actuation rates, and simulators, comprising more than 11,298 motion sequences. Second, we perform payload-aware system identification to isolate payload-related effects and quantify their contributions, and analyze sim-to-real gaps across different simulators. Third, we train the operator **GapONet** to predict delta action for real-time, generalized, payload-conditioned compensation. We further introduce actuation functions and sensor predictors, which enable parallel RL training of **GapONet** with substantially reduced energy consumption. While tracking unseen motions, **GapONet** keeps the incidence of large sim-to-real gaps below 0.09%, whereas competing methods remain near 10%. By correcting upper-body gaps, **GapONet** also stabilizes lower-body locomotion tracking, laying the foundation for improved performance in humanoid loco-manipulation tasks.

## 1 Introduction

Policies trained in simulation benefit from GPU acceleration and massively parallel sampling, enabling fast and scalable optimization under approximate physics such as mass, friction, and damping (Makoviychuk et al., 2021; Tan et al., 2018). However, object interactions in the real world often diverge from these idealizations due to unmodeled or state-dependent effects, most notably in friction, inertia, and contact—leading to a persistent model–plant mismatch (Tobin et al., 2017; Zhao et al., 2020). This sim-to-real gap is further exacerbated in humanoids that manipulate objects of different masses. Variations in end-effector payload induce coupled drifts in equivalent joint inertia, gravity–torque amplitudes via center-of-mass and lever-arm shifts, transmission friction and efficiency, thereby altering closed-loop dynamics (Spong et al., 2006). Yet during policy training, these payload-dependent adjustments are typically simplified or held fixed, which leaves the gap largely unaddressed. The sim-to-real gap can grow in complex, nonpredictive ways, posing a substantial obstacle to robust policy transfer and reliable real-world deployment (Zhang et al., 2023).

Prevailing approaches either calibrate simulators via system identification to tune masses, frictions, and damping (Ljung, 1998; Åström & Eykhoff, 1971; Nelles, 2002); broaden training distributions through domain randomization and observation noise to reduce overfitting (Mehta et al., 2020; Tobin et al., 2017; Chen et al., 2021; Laskey et al., 2017; Zhang et al., 2020; Matas et al., 2018); or stage learning with curricula or progressively harder terrains to harden policies over time (Luo et al., 2020; Wang et al., 2021; Peng et al., 2020; Heess et al., 2017) to bridge the sim-to-real gap. However, the interacted object (payload) is a structured operating condition, not mere noise (Slotine & Li, 1987): it deterministically alters gravity loading, effective inertia, dissipation, and hence the closed-loop gain/phase under PD control. Single-point identification cannot capture behavior across payloads,

and domain randomization or curricula largely treat the payload as unstructured uncertainty. Thus, while these strategies can improve robustness, they hinge on manual design (randomization ranges, noise schedules, curriculum pacing) and provide limited diagnostic attribution. Critically, they do not yield a generalizable representation of the sim-to-real gap for humanoid interaction.

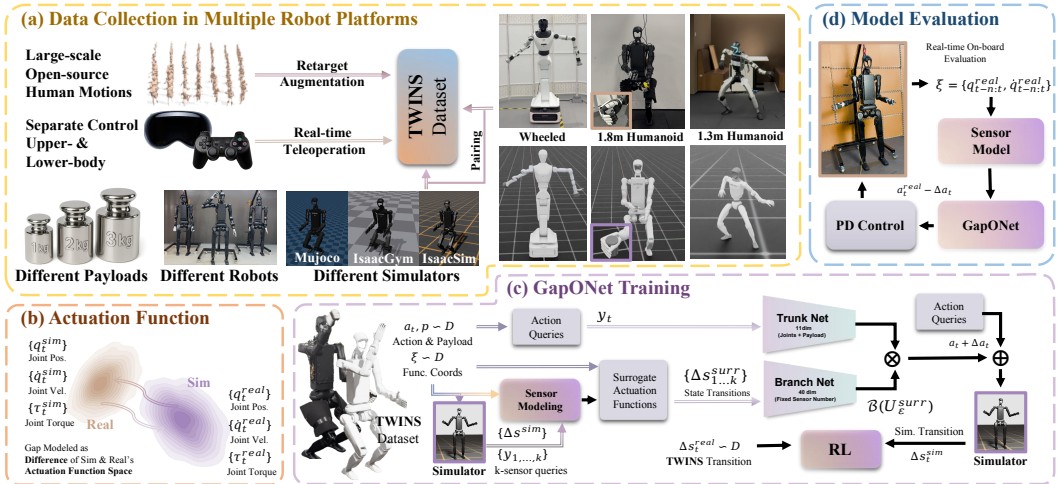

Figure 1: **The overall architecture of both data collection and `GapONet` training.** (a) `TWINS`, a paired sim–real dataset via motion retargeting and real-time teleoperation across diverse payloads, robots, and simulators. (b) The sim–real gap is formulated as a discrepancy between *actuation function* spaces, providing functional coordinates. (c) `GapONet` learns a payload-conditioned nonlinear operator that maps simulation context to residual actions, and training uses parallel RL. (d) Online evaluation on unseen hardware with PD control and sensor modeling to quantify sim–real alignment.

A complementary line of work learns dynamics directly from real data, either as state-transition models or action-to-effect maps (Shi et al., 2019; Xiao et al., 2024; He et al., 2025). From a control standpoint, however, identifying payload-dependent dynamics from passive logs requires persistence of excitation and explicit treatment of operating conditions. In practice, motion patterns, contact regimes, and payload values co-vary, so a single black-box model fit to mixed data tends to entangle payload effects with task-specific artifacts, yielding spurious correlations. As a result, such models often need large volumes of paired sim–real trajectories to cover the space and still exhibit poor cross-payload and unseen-motion generalization. The missing ingredient is a representation that disentangles exogenous operating parameters from state evolution, rather than collapsing them into a single dynamics model. Such a formulation enables a more faithful mapping between the simulator and real-world domains.

We present a ⟨collect–analyze–solve⟩ framework to learn this representation for bridging the sim-to-real gap in humanoids. We first curate `TWINS`, a time-synchronized sim–real corpus with a structured factorial design. Unlike prior collections (Wu et al., 2024; Mao et al., 2024; AgiBot-World-Contributors et al., 2025), our dataset design over diverse payload levels, humanoid platforms, actuation rates, simulations, and motion families, enabling further controlled analyses. To clarify the `GapONet`'s learning target, we first perform gray-box, block-wise system identification atop a PD control model, attributing error reductions to specific payload-related terms and quantifying their contributions. We then analyze identical motions across payloads and simulators, showing structured residuals dominated by actuator nonlinearities, which motivates a more generalizable nonlinear operator rather than a pointwise approximation function.

We then propose `GapONet`, a payload-conditioned nonlinear operator that maps simulation context functions to a residual actions for hardware. Our operator is parameterized with a branch–trunk decomposition (Lu et al., 2019): The branch net encodes the local dynamics of the physical world in which our robot resides as a function, and the trunk network encodes the input variables to that function, including payload weight and target pose. This separation provides a strong structural inductive bias, disentangling the conditioning context from the queried response, thereby enhancing the oper-

ator's generalization capacity. We also propose the sensor predictor, enabling parallel RL training of **GapONet** with lower energy cost while preserving generalization beyond pointwise regression. While tracking unseen motions, **GapONet** keeps the incidence of large sim-to-real gaps below 0.09%, whereas competing methods remain near 10%. By correcting upper-body gaps, **GapONet** also stabilizes lower-body locomotion tracking, laying the foundation for improved performance in humanoid loco-manipulation tasks.

This paper makes three primary contributions:

- We develop a sim-real data collection pipeline and we curate the first dataset **TWINS** focusing on payload-induced sim-real gap across multiple payloads, robots, motions, and simulators.

- We reproduced over 30 hours of real data across four simulators and conducted controlled, ceteris paribus comparisons, yielding quantitative evidence that sim-to-sim evaluation improves the deployability of humanoid controllers.

- We introduce **GapONet**, a payload-conditioned nonlinear operator that maps simulation context functions to residual actions for hardware, and demonstrate its training via RL.

## 2 RELATED WORK

**Sim-to-Real Gap**    Sim-to-real research has largely moved from system identification—calibrating masses, frictions, and control gains to align simulation with measurements (Sobanbabu al al., 2025; Gu et al., 2024; Zhang et al., 2024)—to domain randomization, which perturbs dynamics and observations to harden policies (Peng et al., 2018; Xie et al., 2021; Mehta et al., 2020; Chen et al., 2021). The former can deliver high fidelity but typically demands accurate structural assumptions and extensive hardware time—a challenge that extends not only to classical system identification (Ljung, 1998; Miller et al., 2025) but also to nonlinear methods such as neural-network (Hwangbo et al., 2019; Boussaada et al., 2018; Kuschewski et al., 1993) and kernel-based models (Deisenroth et al., 2013; Zhang et al., 2007), which likewise require substantial data and careful modeling assumptions; the latter proved influential for legged and humanoid control (Xie et al., 2020; Margolis et al., 2024; Li et al., 2023) yet can bias policies toward conservatism (He et al., 2024). In practice, both families often require substantial manual retuning across agents, tasks, and operating regimes, motivating data-driven directions that learn from collected data. One line models actuator nonlinearities with fine granularity to capture motor-level effects (Hwangbo et al., 2019); another emphasizes residual correction, learning delta actions for online compensation with lighter overhead (He et al., 2025). In parallel, simulation–real fusion seeks coverage and speed from simulators while retaining real-world grounding (Fey et al., 2025; Zhang et al., 2023; Bjelonic et al., 2025; Xu et al., 2025), and new benchmarks standardize evaluation (Wu et al., 2024). Despite these advances, both simulator-centric and data-centric pipelines still struggle with broad generalization under real-world variability (Muratore et al., 2022), which limits general gap-bridging in complex systems, such as humanoids.

**Nonlinear Operator Learning**    Operator learning aims to model mappings between function spaces, rather than pointwise input–output relations (Kovachki et al., 2023). In this setting, Unstacked Deep Operator Network (DeepONet) provides a principled architecture with an operator-level universal approximation guarantee (Lu et al., 2019). Its branch–trunk decomposition separately embeds input functions and query variables, yielding a flexible and theoretically grounded representation (Hornik et al., 1989; Lu et al., 2021). Recent work has begun extending operator learning to control and engineering, including Hamilton–Jacobi policy iteration (Lee & Kim, 2025), physics-informed optimal control (Na & Lee, 2024), and operator-based model-predictive control (de Jong et al., 2025). Beyond control, multiphysics applications demonstrate operator surrogates for solution fields in materials processing and additive manufacturing, highlighting scalability to complex PDE-governed phenomena (Kushwaha et al., 2024). However, these efforts remain largely theory-driven or tailored to specific domains, with limited focus on robotics sim-to-real—especially for humanoids operating under shifting payload-dependent dynamics. This gap calls for an operator-based formulation that can explicitly condition on task and environment variations and learn the functional discrepancies between simulation and reality, while preserving sample efficiency and real-time applicability.

# 3 DATA COLLECTION AND GAP ANALYSIS

End-effector payloads reshape joint dynamics and closed-loop behavior—raising reflected inertia, shifting gravity torques, and coupling with actuator and contact nonlinearities. Divergent simulator treatments of these effects produce a persistent, multi-factor sim-to-real gap. This section provides a structured diagnosis: Section 3.1 isolates payload-induced terms via gray-box system identification; Section 3.3 compares simulators on identical payload-bearing motions under matched controllers; Section 3.2 details **TWINS** and its collection pipeline.

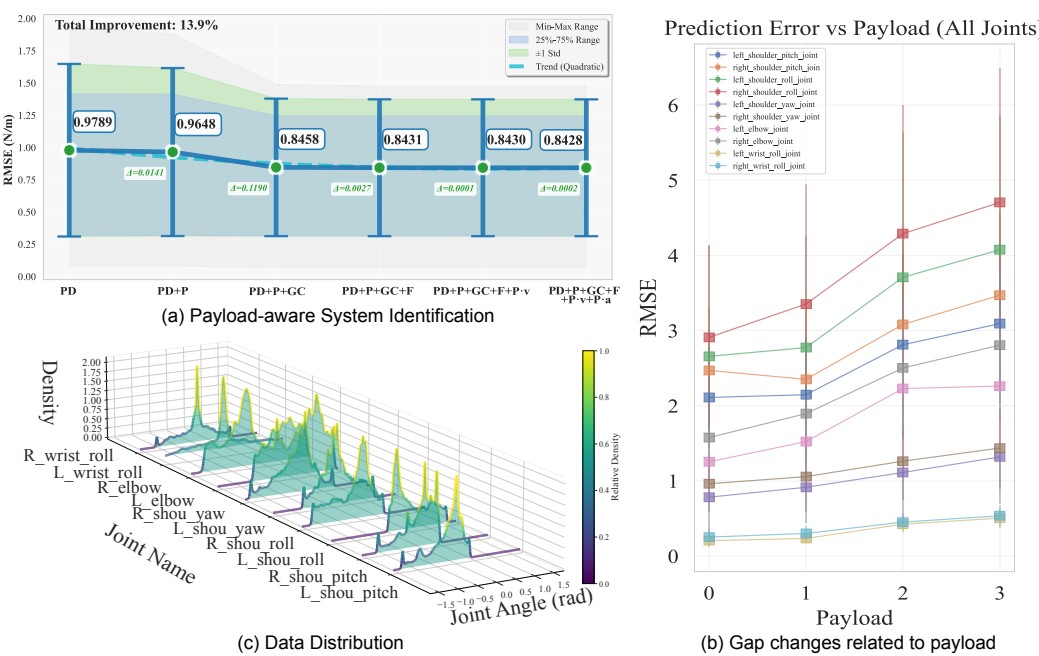

Figure 2: **System identification and data distribution** (a) Prediction residuals after adding payload-related parameters; notably, adding gravity compensation yields a clear improvement. (b) The vertical axis shows the change in the joint-wise gap as the payload increases. (c) Data distribution of **TWINS**; the z-axis indicates the probability density of each joint action.

## 3.1 PAYLOAD-AWARE SYSTEM IDENTIFICATION

Using bipedal humanoids that demand precise control as exemplars (Unitree H1-2 and G1), both operate under joint-space PD control tailored to locomotion (details in Section A.4.1). With added end-effector payloads $P$, we adopt a gray-box identification scheme: start from a rigid PD baseline and progressively augment the torque model with physically grounded terms salient in manipulation. For each joint, we fit a linear in parameters regression that attributes the sim-to-real discrepancy to gravity scaling, reflected inertia, actuator and transmission nonlinearities, and contact compliance, and we quantify their marginal contributions:

$$
\begin{aligned}
\tau = {} & K_p \left( q_{\mathrm{cmd}} - q \right) + K_d \left( \dot{q}_{\mathrm{cmd}} - \dot{q} \right) + K_v \, \dot{q} + K_c \tanh\!\left( \frac{\dot{q}}{\varepsilon} \right) \\
& + K_{\mathrm{payload}} \, P \\
& + K_{P\sin} \, P \, \sin q + K_{P\cos} \, P \, \cos q \\
& + K_{P\dot{q}} \, P \, \dot{q} + K_{P\ddot{q}} \, P \, \ddot{q} \\
& + \tau_0.
\end{aligned}
\tag{1}
$$

Here, $K_p$ and $K_d$ are proportional and derivative gains; $K_v$ and $K_c$ model viscous and Coulomb friction with $\varepsilon$ smoothing the latter; $K_{\mathrm{payload}}$ scales the main payload $P$; $K_{P\sin}$ and $K_{P\cos}$ capture gravity and posture coupling under payload; $K_{P\dot{q}}$ and $K_{P\ddot{q}}$ model interactions between payload and

joint velocity or acceleration; $\tau_0$ is a constant bias. The remaining symbols are $\tau$ for joint torque; $q, \dot{q}, \ddot{q}$ for joint position, velocity, and acceleration; $q_{\mathrm{cmd}}, \dot{q}_{\mathrm{cmd}}$ for commanded references; and $P$ for payload magnitude interpreted as mass or equivalent inertia at the end effector. All $K$ coefficients are identified per joint. This compact form separates baseline PD, friction, and payload dependent effects and enables clear attribution of simulation to real error.

Using over 2,000 data collected from real robots, we fit Equation (1) by minimizing $RMSE$ between its torque and measurements. Adding payload-dependent terms reduces error Figure 2(a), with gravity compensation giving an early gain, but at higher payloads Equation (1) no longer captures the closed loop response Figure 2(b). The equation is not a replica of the simulator; it is a control equivalent surrogate that covers dominant channels under matched controllers. Identification on synchronized inputs with persistently exciting motions enables term level attribution, and the residual exposes nonlinear dynamics not captured by compact models. Learning a nonlinear operator, rather than a pointwise nonlinear function, better supports generalization across trajectories, payload schedules, actuation rates, and robots by mapping context functions to control signals.

### 3.2 **TWINS** COLLECTION

Section 3.1 shows with block-wise identification that the prediction to measurement gap is nonlinear and uncertain. Given the lack of suitable data, to validate this conclusion on genuine sim to real pairs, we present **TWINS**, the first dataset focused on payload induced sim to real gaps across multiple robots, standardized payload levels, and motion classes. **TWINS** records humanoid dynamics hierarchically, from single joints to full upper body motions with 3 different low-body gaits, using four Unitree H1-2 units with end effector masses from 0 to 3 kg (standard calibration weights) and actuation rates of 50 Hz and 100 Hz. The real data totals 30.17 hours, 11,298 sequences, and 307,273 synchronized frames. The distribution appears in Figure 2(c).

Each sequence is time synchronized with a matched high fidelity simulation replica in three widely used humanoid training simulators (MuJoCo, Isaac Gym, Isaac Sim), enabling comparison of real and simulated executions at the frame level and yielding a fourfold paired corpus of 120.68 (one real trace plus three simulated replicas). For every frame we record joint positions $q_{\mathrm{sim}}, q_{\mathrm{real}}$, velocities $\dot{q}_{\mathrm{sim}}, \dot{q}_{\mathrm{real}}$, accelerations $\ddot{q}_{\mathrm{sim}}, \ddot{q}_{\mathrm{real}}$, torques $\tau_{\mathrm{sim}}, \tau_{\mathrm{real}}$, payload $P$, and motor temperature $T_{\mathrm{real}}$. Further details of our collection pipeline and dataset on different robots are in Section A.3.

### 3.3 SIM-TO-REAL GAP ANALYSIS

After post-processing the paired data **TWINS**, we conduct a targeted analysis of the sim-to-real gap to guide operator design for payload-induced nonlinearities. The analysis tests concordance with the block wise identification in Section 3.1, determines whether the effect is concentrated in the upper body or extends to the whole body, and quantifies differences across simulators when reproducing the same motion under matched control.

**Same motion with different lower-body gaits**  We execute 17 upper-body motion sequences under three lower-body conditions: bipedal locomotion, static squat, and stance support only. As shown in Figure 3(a), the outer ellipse marks the shared kinematic envelope, while the center trajectory is the PCA trace of a single motion; across gaits, this trace is nearly retraced with only small phase/offset shifts. With envelopes matched, the upper-body sim-to-real gap is therefore largely insensitive to the lower-body condition, and residual differences are dominated by payload-amplified channels. We quantify this via joint-wise normalized $RMSE$, commanded–measured phase lag, and torque-saturation incidence. Note that, unlike fixed-base dual-arm platforms, upper-body actions in humanoids couple back to locomotion and can stress the gait controller; full experiments and analysis are in Section 5.2.

**Same motion with different payloads**  As shown in Figure 3(b), each colored trajectory plots the joint-wise sim–real residual over time. Increasing payload amplifies both residual magnitude and phase lag, yielding larger state gaps and longer delays. Across **TWINS**, payload consistently widens the gap, and the residual grows nonlinearly with payload mass Figure 2(b), in line with the block-wise identification trends reported in Section 3.1.

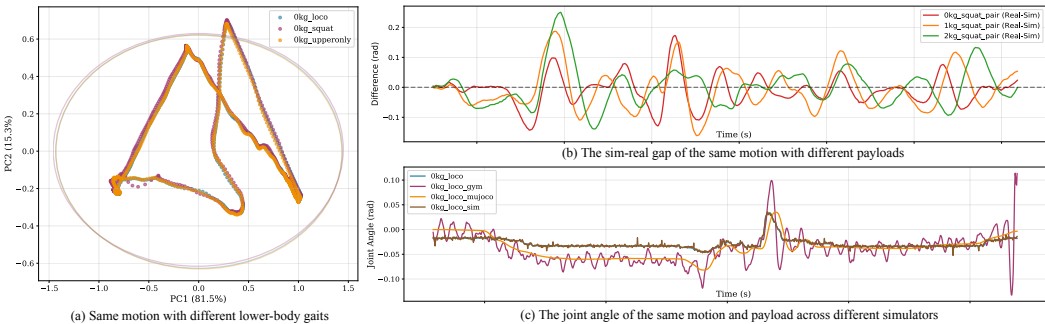

Figure 3: **Gap Analysis.** (a) The outer ellipse marks a shared kinematic envelope across gaits, while the central PCA trajectory of a single motion shows only minor variations with overall consistency. (b) Payload-induced sim-to-real deviation during a squat posture, showing an increasing gap even in a quasi-static state. (c) Joint-angle discrepancies across simulators (Mujuco, IssacGym, IsaacSim) during locomotion, indicating a persistent gap under dynamic motion.

**Same motion across different simulators**   Current methods always apply sim-to-sim evaluation as the cross-validation before hardware deployment (He et al., 2025; Liu et al., 2024). To characterize simulator-specific differences and their dependence on payload, we compare identical motions across MuJoCo, Isaac Gym, and Isaac Sim under matched controllers and simulator-adapted generic parameters over a standardized payload grid. Experiments Figure 3(c) show that MuJoCo yields smoother trajectories but larger peaks in high-acceleration segments; Isaac Gym exhibits occasional joint-level jitter; Isaac Sim achieves the most stable alignment in our evaluations, but still leaves a nonlinear gap during interaction. To stay aligned with prevailing practice and minimize simulator-induced confounds, we adopt Isaac Sim for subsequent experiments, as it exhibits the smallest sim-to-real gap in our analysis. We also release paired data for MuJoCo and Isaac Gym to enable cross-simulator comparisons and support future research. More results in Section A.4.

In summary, across payload levels, all simulators show a nonlinear increase in error relative to real hardware, with simulator-specific modes. This pattern persists across lower-body gaits: when kinematic envelopes are matched, the distributions of upper-body error and phase metrics remain closely aligned. The discrepancy arises from coupled channels—gravity, friction, Coriolis and inertial coupling, actuator limits and efficiency drift, sensing noise, and delays—that a pointwise function mapping cannot capture or generalize. A nonlinear operator is better suited: `GapONet` provides a compact, transferable representation by mapping context functions to corrective control signals across trajectories, payload schedules, actuation rates, and robot morphologies.

## 4    METHOD

We propose `GapONet`, a payload-conditioned nonlinear operator that maps simulation context functions to a residual action for hardware. `GapONet` learns a functional correspondence from simulator space to real dynamics and introduces **actuation functions** that encode command and feedback histories. We then propose the **sensor predictor**, which enables parallel RL training of `GapONet`, overcoming the high energy consumption of the original approach while maintaining generalization beyond pointwise regression.

### 4.1    PROBLEM FORMULATION

Previous methods lack an explicit model of both the simulator and the real world (Mehta et al., 2020; Tobin et al., 2017; Matas et al., 2018; Shi et al., 2019; Xiao et al., 2024; He et al., 2025), thereby limiting their capacity to characterize both domains and constraining the achievable degree of alignment between them. We therefore propose **actuation functions**, which formally model robot actuation in both simulation and reality as **functions**. This approach thereby converts the problem of modeling their discrepancies into one of finding a mapping between their respective function spaces.

These functions characterize the mapping from actions (together with task-specific parameters) to state transitions, under different joint configurations and dynamics, both in simulation and on the real robot.

Formally, bridging the sim-to-real gap can be posed as learning an operator that maps $\mathcal{U}^{\text{sim}}$ to $\mathcal{U}^{\text{real}}$ rather than approximating multiple collected dynamics, where $\mathcal{U}$ denotes the underlying function space. Each actuation function $U \in \mathcal{U}$—the family of actuation functions available to the system—is parameterized by a natural coordinate $\xi$, which encodes the instantaneous joint dynamics determined by the system's current state and joint configuration. Accordingly, our actuation function is defined as $U_\xi : A \times P \to Q \times V$, where $A$, $P$, $Q$, and $V$ denote the space of action, payload, joint position, and joint velocity, respectively. The goal of **GapONet** is to learn an operator $\mathcal{G}$ that aligns the discrepant humanoid motion distributions of simulation and the real world, i.e., $\mathcal{G}(U_\xi^{\text{sim}}) \approx U_\xi^{\text{real}}$ by producing residual actions.

## 4.2 NETWORK STRUCTURE

To effectively learn the operator, we adopt a DeepONet-style architecture (Lu et al., 2019). In this framework, the input function is represented by its values at $k$ fixed sensor locations, which are encoded by the Branch Network; the Trunk Network embeds the query coordinates, and the operator output is obtained via their multiplicative fusion. This design provides a principled way to approximate nonlinear operators by separating the representation of the input function (via the Branch Net) from the evaluation coordinates (via the Trunk Net). The rationale for adopting DeepONet, along with a detailed discussion of its applicability to our problem setting, is provided in Section A.6. All formal notation and value-space definitions are consolidated in Section A.6.2 for reference.

The value of $k$ fixed locations are denoted as $\{x_i\}_{i=1}^{k}$ where $x_i = (a, p)$, with $a \in A$ and $p \in P$ as defined in Section 4.1. More details are in Section A.8. For each location $x_1, \ldots, x_k$, we first query the simulated actuation function $U_\xi^{\text{sim}}$ to obtain sensor readings $S_i$:

$$S_i(U_\xi^{\text{sim}}) = U_\xi^{\text{sim}}(x_i) = \Delta f^{\text{sim}}(s_{\text{sim}}^\xi, x_i), \quad i = 1, \ldots, k, \qquad (2)$$

where $\Delta f^{\text{sim}}$ denotes the simulator's one-step update. $S(U_\xi^{\text{sim}}) = [S_1(U_\xi^{\text{sim}}), \ldots, S_k(U_\xi^{\text{sim}})]$ denotes the concatenation of the $k$ sensor values, providing a structured representation of the actuation function. $S(U_\xi^{\text{sim}})$ is then embedded into a latent representation via the Branch Net $\mathcal{B}$:

$$\mathcal{B}(U_\xi^{\text{sim}}) = [\mathcal{B}_1(S(U_\xi^{\text{sim}})), \ldots, \mathcal{B}_n(S(U_\xi^{\text{sim}}))], \qquad (3)$$

where $n$ denotes the number of branch features, with each $\mathcal{B}_i$ encoding a distinct feature of the actuation function parameterized by the natural coordinates $\xi$, decomposing complex dynamics into interpretable subcomponents.

The Trunk Net $\mathcal{T}$ encodes query signals that combine the payload and the current-timestep action:

$$y \in A \times P, \quad \mathcal{T}(y) = [\mathcal{T}_1(y), \ldots, \mathcal{T}_n(y)], \qquad (4)$$

where the trunk features share the same dimension $n$ as the branch features. We then define the operator $G_\theta(\xi, y)$ by fusing the Branch output $\mathcal{B}(U_\xi^{\text{sim}})$ and the Trunk output $\mathcal{T}(y)$ through an element-wise product:

$$G_\theta(\xi, y) = \mathcal{B}(U_\xi^{\text{sim}}) \odot \mathcal{T}(y) = \left[ \mathcal{B}_1(S(U_\xi^{\text{sim}})) \cdot \mathcal{T}_1(y), \ldots, \mathcal{B}_n(S(U_\xi^{\text{sim}})) \cdot \mathcal{T}_n(y) \right], \qquad (5)$$

where each trunk feature $\mathcal{T}_j$ encodes the input queries in the coordinate system defined by the basis output from the corresponding branch feature $\mathcal{B}_j$.

Inspired by residual dynamics modeling (He et al., 2025), we do not directly supervise the operator output $G_\theta(\xi, y)$ using data from **TWINS**. Instead, **GapONet** predicts a per-joint corrective delta action, which is applied on top of the simulator's nominal command. In this view, $G_\theta$ produces the residual action needed to compensate for the mismatch between simulation and reality. The resulting operator is defined as:

$$\mathcal{G}(U_\xi^{\text{sim}})(y_t) = \Delta f^{\text{sim}} \left( s_{\text{sim}}^\xi, a_t + G_\theta(\xi, y_t) \right). \qquad (6)$$

### 4.3 GPU-PARALLEL OPERATOR LEARNING

Training an operator to generate physically consistent delta actions is challenging, as it requires real-time evaluations of a non-differentiable simulator and repeated computation of sensor values for every actuation coordinate $\xi$. These constraints preclude direct supervised learning, motivating our use of reinforcement learning (explained in Section A.6.3). To further improve efficiency, we introduce a **sensor model** $S_\phi$ that predicts sensor readings from near-history dynamics $h$, approximating the output of the actuation function parameterized by $\xi$:

$$\mathcal{L}_{\text{sensor}} = \mathbb{E}_\xi \left[ \sum_i \left\| \Delta f^{\text{sim}}(s_{\text{sim}}^\xi, x_i) - (S_\phi(h))_i \right\|_2^2 \right]. \tag{7}$$

Optimizing $\phi$ yields a surrogate function space $\mathcal{U}^{\text{surr}} = S_\phi(\mathcal{U}^{\text{sim}})$, where $S_\phi$ maps each simulated actuation function $U_\xi^{\text{sim}}$ to a smooth, computationally lightweight surrogate $U_h^{\text{surr}}$ with matching sensor behavior. This surrogate space replaces the expensive and non-differentiable simulator-based function space $\mathcal{U}^{\text{sim}}$ with one that is differentiable, easy to sample, and amenable to large-scale GPU-parallel training. As a result, learning the sim-to-real operator becomes a tractable problem of mapping $\mathcal{U}^{\text{surr}}$ to $\mathcal{U}^{\text{real}}$. We denote by $\mathcal{D}$ the **TWINS** dataset distribution over all collected tuples $(h, \xi, y)$ used for training, which gives rise to the following objective:

$$\underset{\theta}{\text{minimize}} \ \mathbb{E}_{h,\xi,y \sim \mathcal{D}} \left[ \left\| \mathcal{G}(U_h^{\text{surr}})(y) - U_\xi^{\text{real}}(y) \right\|_2^2 \right]. \tag{8}$$

This objective minimizes the functional discrepancy between the surrogate and real actuation functions. It can be equivalently expressed as a reinforcement-learning problem with the reward:

$$r_t = -w \left\| (s_{\text{real}}^{t+1} - s_{\text{real}}^t) - \mathcal{G}(U_h^{\text{surr}})(y_t) \right\|_2^2, \tag{9}$$

where $s_{\text{real}}$ and $y$ are sampled from $\mathcal{D}$. Maximizing the expected episodic reward under this reward function aligns with Equation (8). In practice, we optimize $\theta$ with PPO Schulman et al. (2017). The operator $G_\theta$ is trained as a stochastic policy, defined by $G_\theta(\cdot) + \mathcal{N}(\mathbf{0}; \sigma\mathbf{I})$, where $\sigma$ is a learnable parameter that gradually decays to zero during training. We adopt the standard clipped surrogate objective:

$$\mathcal{L}_{\text{PPO}}(\theta) = -\mathbb{E}_t[\min\left(r_t(\theta)A_t, \ \text{clip}(r_t(\theta), 1 - \epsilon, 1 + \epsilon) \, A_t\right)], \tag{10}$$

where $r_t(\theta)$ denotes importance sampling ratio, and $A_t$ is the advantage computed from the reward $r_t$ in Equation (9). This yields stable updates under non-differentiable dynamics. We provide pseudo code for the training algorithm in Algorithm 1.

## 5 EXPERIMENT

Our experimental evaluation comprises two parts: Section 5.1 evaluates **GapONet**'s zero-shot generalization to unseen robots and motions; Section 5.2 measures improvements in humanoid locomotion stability through online residual compensation on hardware.

### 5.1 ZERO-SHOT MOTION TRACKING

**GapONet** can generalize to unseen target joint-position sequence (motion) under the branch–trunk architecture. To test this capability beyond our dataset **TWINS**, we collected an unseen-motion test set of 100 sim–real pairs: 35 sequences at 0 kg, 23 at 1 kg, 22 at 2 kg, and 20 at 3 kg. These data are intentionally kept out of the training set in order to further test the model's generalization performance on unseen payload conditions. The test set also spans three lower-body gaits in a 6:3:1 ratio for static stance, squat, and locomotion. For quantitative assessment, we report **Large Gap Ratio** (the percentage of frames whose error exceeds a predefined threshold), **IQR** (the interquartile range of the gap over all motions), and **Gap Range** (the framewise gap range from minimum to maximum).

In this motion tracking setting, the trained **GapONet** takes simulator-side inputs (action, payload, joint position, and joint velocity) to produce a corrective $a_{\text{sim}}^t + \Delta a_t$, which is added to the simulator's command to obtain $s_{\text{sim}}^{t+1}$ and compared against time-synchronized real measurements $s_{\text{real}}^{t+1}$. We

benchmark `GapONet` with four baselines: (i) an MLP learned dynamics model (He et al., 2025), (ii) a Transformer learned dynamics model that exploits temporal context better, (iii) system identification, a classical approach to bridging the sim-to-real gap, and (iv) PD control with official gains. Each experiment is repeated multiple times, and we report the mean and standard deviation in the table. As shown in Table 1, `GapONet` attains the best or tied-best scores on nearly all metrics, with a pronounced improvement in LGR. These results indicate smoother, more controllable zero-shot gap bridging than the learned dynamics baselines and consistent gains over system identification across motions from multiple robots.

Table 1: Zero-shot sim-to-real gap on unseen-motion test set across four payloads.

| Method | 0 kg | | | 1 kg | | |
|---|---|---|---|---|---|---|
| | LGR(%) ($\downarrow$) | IQR ($\downarrow$) | Range ($\downarrow$) | LGR(%) ($\downarrow$) | IQR ($\downarrow$) | Range ($\downarrow$) |
| PD control | $12.7^{\pm 3.3}$ | $0.138^{\pm 0.007}$ | $0.538^{\pm 0.019}$ | $10.6^{\pm 0.1}$ | $0.139^{\pm 0.028}$ | $0.667^{\pm 0.011}$ |
| MLP | $10.0^{\pm 0.8}$ | $0.108^{\pm 0.012}$ | $0.480^{\pm 0.088}$ | $10.8^{\pm 0.1}$ | $0.125^{\pm 0.002}$ | $0.589^{\pm 0.029}$ |
| Transformer | $9.55^{\pm 0.3}$ | $0.127^{\pm 0.014}$ | $0.465^{\pm 0.067}$ | $5.60^{\pm 0.4}$ | $0.140^{\pm 0.005}$ | $\mathbf{0.525}^{\pm 0.041}$ |
| Domain Randomization | $3.17^{\pm 0.6}$ | $0.119^{\pm 0.010}$ | $0.548^{\pm 0.066}$ | - | - | - |
| System Identification | $12.4^{\pm 0.3}$ | $0.141^{\pm 0.015}$ | $0.505^{\pm 0.032}$ | $9.01^{\pm 1.0}$ | $0.140^{\pm 0.029}$ | $0.609^{\pm 0.122}$ |
| Network-based SysID | $12.5^{\pm 0.06}$ | $0.154^{\pm 0.019}$ | $0.441^{\pm 0.001}$ | $13.1^{\pm 0.65}$ | $0.129^{\pm 0.031}$ | $0.538^{\pm 0.002}$ |
| Kernel-based SysID | $13.3^{\pm 0.14}$ | $0.155^{\pm 0.019}$ | $0.497^{\pm 0.006}$ | $8.84^{\pm 2.37}$ | $0.129^{\pm 0.015}$ | $0.588^{\pm 0.002}$ |
| **GapONet** (Ours) | $\mathbf{0.09}^{\pm 0.03}$ | $\mathbf{0.093}^{\pm 0.016}$ | $\mathbf{0.449}^{\pm 0.117}$ | $\mathbf{0.22}^{\pm 0.11}$ | $\mathbf{0.115}^{\pm 0.013}$ | $0.537^{\pm 0.148}$ |

| Method | 2 kg | | | 3 kg | | |
|---|---|---|---|---|---|---|
| | LGR(%) ($\downarrow$) | IQR ($\downarrow$) | Range ($\downarrow$) | LGR(%) ($\downarrow$) | IQR ($\downarrow$) | Range ($\downarrow$) |
| PD control | $11.2^{\pm 0.1}$ | $0.205^{\pm 0.001}$ | $0.625^{\pm 0.038}$ | $12.8^{\pm 0.1}$ | $0.499^{\pm 0.008}$ | $0.642^{\pm 0.060}$ |
| MLP | $10.8^{\pm 0.1}$ | $0.252^{\pm 0.003}$ | $0.621^{\pm 0.023}$ | $12.2^{\pm 0.9}$ | $0.460^{\pm 0.013}$ | $0.668^{\pm 0.060}$ |
| Transformer | $0.44^{\pm 0.3}$ | $\mathbf{0.140}^{\pm 0.002}$ | $0.606^{\pm 0.040}$ | $9.82^{\pm 0.1}$ | $0.416^{\pm 0.002}$ | $0.573^{\pm 0.178}$ |
| Domain Randomization | - | - | - | - | - | - |
| System Identification | $9.53^{\pm 0.7}$ | $0.193^{\pm 0.102}$ | $0.601^{\pm 0.031}$ | $12.1^{\pm 0.5}$ | $0.494^{\pm 0.003}$ | $0.611^{\pm 0.127}$ |
| Network-based SysID | $12.8^{\pm 0.05}$ | $0.198^{\pm 0.001}$ | $0.609^{\pm 0.001}$ | $12.5^{\pm 0.5}$ | $0.415^{\pm 0.074}$ | $0.626^{\pm 0.074}$ |
| Kernel-based SysID | $8.88^{\pm 1.23}$ | $0.183^{\pm 0.001}$ | $0.618^{\pm 0.005}$ | $8.45^{\pm 0.06}$ | $0.478^{\pm 0.075}$ | $0.605^{\pm 0.51}$ |
| **GapONet** (Ours) | $\mathbf{0.39}^{\pm 0.10}$ | $0.161^{\pm 0.004}$ | $\mathbf{0.578}^{\pm 0.112}$ | $\mathbf{0.84}^{\pm 0.23}$ | $\mathbf{0.317}^{\pm 0.005}$ | $\mathbf{0.498}^{\pm 0.157}$ |

## 5.2 Locomotion Trajectory Tracking

Section 5.1 demonstrates the generalization and gap-solving capabilities of `GapONet`, but improving upper-body tracking alone is insufficient to prove system-level benefits. For broader humanoid applications, lower-body motion must also be considered. As shown in Section 3.3, lower-body gaits have minimal impact on upper-body motion distributions, while upper-body compensation affects the lower-body dynamics through coupled torques and contact forces, influencing the center of mass trajectory (Zhang et al., 2025). Motivated by this asymmetry, we further evaluate `GapONet`'s ability to preserve lower-body locomotion stability by correcting upper-body discrepancies. To this end, we deploy `GapONet` as an online residual compensator on hardware, enabling it to refine upper-body actions in real time and thereby improve lower-body dynamics during locomotion. At each time step, `GapONet` receives real-side inputs (action, payload, joint position, and joint velocity) and predicts a corrective term ($\Delta a_t$). The executed command is then computed as $a'_t = a^t_{\text{real}} - \Delta a_t$, which is applied to the robot to obtain the next real state $s^{t+1}_{\text{real}}$. This state is compared against the time-aligned simulated state $s^{t+1}_{\text{sim}}$. More details in Section A.7.6.

We provide both qualitative and quantitative results to evaluate the performance of `GapONet`. We conducted tests on 14 motion sequences (7 at 0 kg and 7 at 1 kg payloads) using a previously unseen Unitree H1-2 robot. For quantitative assessment, we report **Trajectory Consistency** (velocity discrepancy between simulation and real data), **Smoothness** (mean acceleration gap), and **Robustness** (per-joint gap with added noise). Each experiment was repeated multiple times, and the results are

Table 2: Sim-to-real gap in locomotion trajectory tracking on an unseen humanoid robot.

| Method | Trajectory Consistency ($\downarrow$) | | Smoothness ($\downarrow$) | | Robustness ($\downarrow$) | |
|---|---|---|---|---|---|---|
| | 0 kg | 1 kg | 0 kg | 1 kg | 0 kg | 1 kg |
| PD control | $20.33^{\pm 1.982}$ | $27.49^{\pm 1.057}$ | $53.76^{\pm 0.257}$ | $25.76^{\pm 0.277}$ | $10.16^{\pm 0.007}$ | $\mathbf{10.14}^{\pm 0.026}$ |
| MLP | $19.18^{\pm 0.919}$ | $28.82^{\pm 1.560}$ | $53.48^{\pm 0.343}$ | $25.55^{\pm 0.361}$ | $10.15^{\pm 0.027}$ | $\mathbf{10.14}^{\pm 0.024}$ |
| Transformer | $19.13^{\pm 0.689}$ | $29.05^{\pm 1.576}$ | $53.57^{\pm 0.290}$ | $26.56^{\pm 0.385}$ | $10.14^{\pm 0.007}$ | $10.16^{\pm 0.012}$ |
| System Identification | $19.16^{\pm 0.489}$ | $28.59^{\pm 1.343}$ | $\mathbf{24.99}^{\pm 0.298}$ | $25.16^{\pm 0.378}$ | $10.14^{\pm 0.011}$ | $10.17^{\pm 0.008}$ |
| **GapONet** (Ours) | $\mathbf{18.78}^{\pm 1.147}$ | $\mathbf{23.23}^{\pm 5.245}$ | $53.36^{\pm 0.486}$ | $\mathbf{25.08}^{\pm 0.181}$ | $\mathbf{10.13}^{\pm 0.167}$ | $\mathbf{10.14}^{\pm 0.017}$ |

Values are reported as mean with superscript $\pm$ standard deviation (three decimals). The best result in each column is highlighted in light green and bold.

presented as mean and standard deviation to ensure validity. Detailed metric calculations can be found in Section A.7.

Results in Table 2 show that **GapONet** outperforms other methods in trajectory tracking, maintaining excellent performance even with payloads, and exhibiting the smallest error growth. In qualitative analysis, as shown in Figure 4, when a humanoid robot follows the same trajectory from the same starting point with identical commands, the real execution trajectory (depicted by the white lines) exhibits significant deviations. Robots without the residual model show frequent tilting and large trajectory shifts, while the policy with **GapONet** follows better. Full video demonstrations and more details can be found in Section A.7 and the supplementary material.

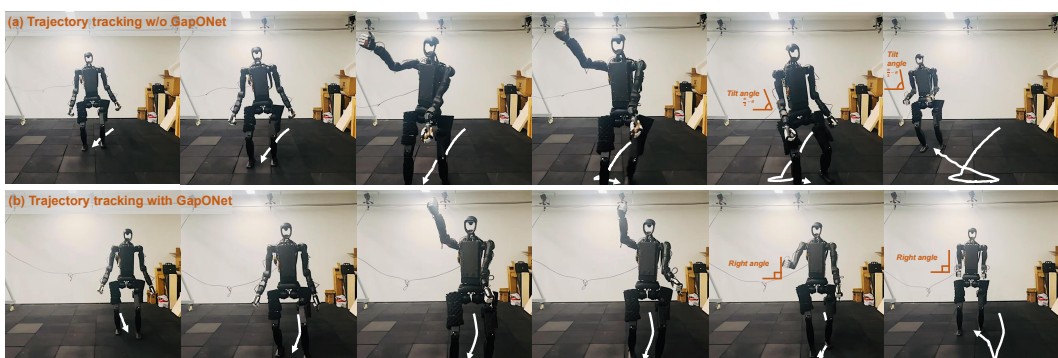

Figure 4: **Locomotion trajectory tracking.** (a) shows trajectory tracking using PD control, where the path (white line) deviates significantly, and the robot's torso tilts drastically, indicating instability. (b) shows the full-body motion after upper-body correction with **GapONet**. Although there is still some rightward deviation, the trajectory is much more stable, and the robot's torso remains upright.

These results collectively demonstrate the generalization and gap-solving capabilities of **GapONet**. It not only outperforms current baselines on unseen motions under different payloads but also achieves higher stability in lower-body locomotion on an unseen robot, laying the foundation for improved performance in humanoid loco-manipulation tasks.

## 6 CONCLUSION

We present an end-to-end data-collection pipeline and curate 120+ hours of paired sim–real data across multiple robots. We characterize payload-related parameters, compare sim-to-real gaps across simulators, and assess the impact of lower-body actions on whole-body behavior. We then learn a payload-conditioned nonlinear operator **GapONet** mapping simulation context functions to residual actions for hardware. On zero-shot motion tracking, the large-gap ratio is 0.09%, with improved robustness and smoothness in locomotion trajectory tracking, strengthening the basis for humanoid loco-manipulation. Future work and limitations are discussed in Section A.9.

ETHICS STATEMENT

The dataset used and planned for release in this work has been fully anonymized and does not contain any personal or individually identifiable information, but rather consists of a collection of publicly accessible content. The paper does not include any analysis, reporting, or disclosure of private user details, and care has been taken to ensure that all data handling aligns with privacy regulations and ethical guidelines.

REPRODUCIBILITY STATEMENT

We include real-world experimental footage to substantiate the reported results and release a subset of sim–real paired data for cross-validation; both are provided in the supplementary materials. Key implementation details and experimental settings are described in the main paper (Section 4, Section 5) and supplementary materials Section A.8.

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

# A  APPENDIX

## A.1  THE USE OF LARGE LANGUAGE MODELS (LLMS)

We employed LLMs only for grammar/style rewrites and equation/notation formatting corrections. We appreciate the steadily improving reasoning capabilities of LLMs, which helped us identify linguistic issues more quickly and maintain a more consistent scholarly style. However, all research ideation, theoretical development and formula derivations, methodological choices, and experimental design and execution were performed exclusively by the authors. Accordingly, the LLM did not play a significant role in research ideation or writing and should not be regarded as a contributor.

## A.2  OPEN-SOURCE RELEASE

To support reproducibility and foster further research on humanoid sim-to-real transfer, we will release the full codebase, training pipelines, pretrained **GapONet** models, and the complete **TWINS** dataset upon publication. The release includes (i) data collection and synchronization tools for paired sim–real recording across payloads, robots, and simulators, (ii) operator-learning implementations with DeepONet-based architectures, (iii) reinforcement learning pipelines with surrogate actuation functions and sensor predictors, and (iv) evaluation scripts for sim-to-sim and sim-to-real benchmarking. All resources will be made publicly available under a permissive license, enabling the community to build upon our framework, reproduce all experimental results, and extend the dataset for broader loco-manipulation tasks.

## A.3  DATA COLLECTION

### A.3.1  LEGGED HUMANOID ROBOT

We collect paired sim–real data on two humanoids: the 1.8 m Unitree H1-2 and the 1.3 m Unitree G1. Joint naming and kinematic locations are shown in Figure 5. In our setup, we log the full upper body and locomotion-relevant joints (27-DoF configuration in code), along with IMU and actuator telemetry.

**ROS setup and topics**  Data acquisition is implemented as a ROS 2 Python node (`rclpy`, node name `deploy_node`). The node subscribes to low-level robot state messages and publishes torque/position commands:

- *Subscriptions:* `LowState` (joint positions/velocities/currents, IMU, wireless remote), used to buffer sensor streams and teleop events.
- *Publications:* `LowCmd` on topic `lowcmd_buffer` at 50 Hz (control period $\Delta t \approx 20\,\mathrm{ms}$). Commands include per-joint PD terms and optional feedforward residuals (CRC is appended before transmission).

Teleoperation triggers (e.g., start/stop, emergency stop) are parsed from the wireless controller and gate recording and command streaming.

**What is recorded**  For each trial, we write files (per-trial timestamped) with the following datasets, matching the code:

- `command_time_list` (s): wall-clock times when commands are produced.
- `command_val_list`: commanded action vectors (per 20 ms tick).
- `robot/joint_time_list` (s): time stamps associated with the sensed robot state.
- `robot/joint_angle_list`, `robot/joint_velocity_list`, `robot/joint_current_list`, `robot/joint_temperature_list`: actuator telemetry.
- `robot/imu_list`, `robot/ang_vel_list`: IMU linear orientation proxies and angular rates.
- `motion_name`, `current_time`: metadata for the retargeted/teleop motion and file creation time.

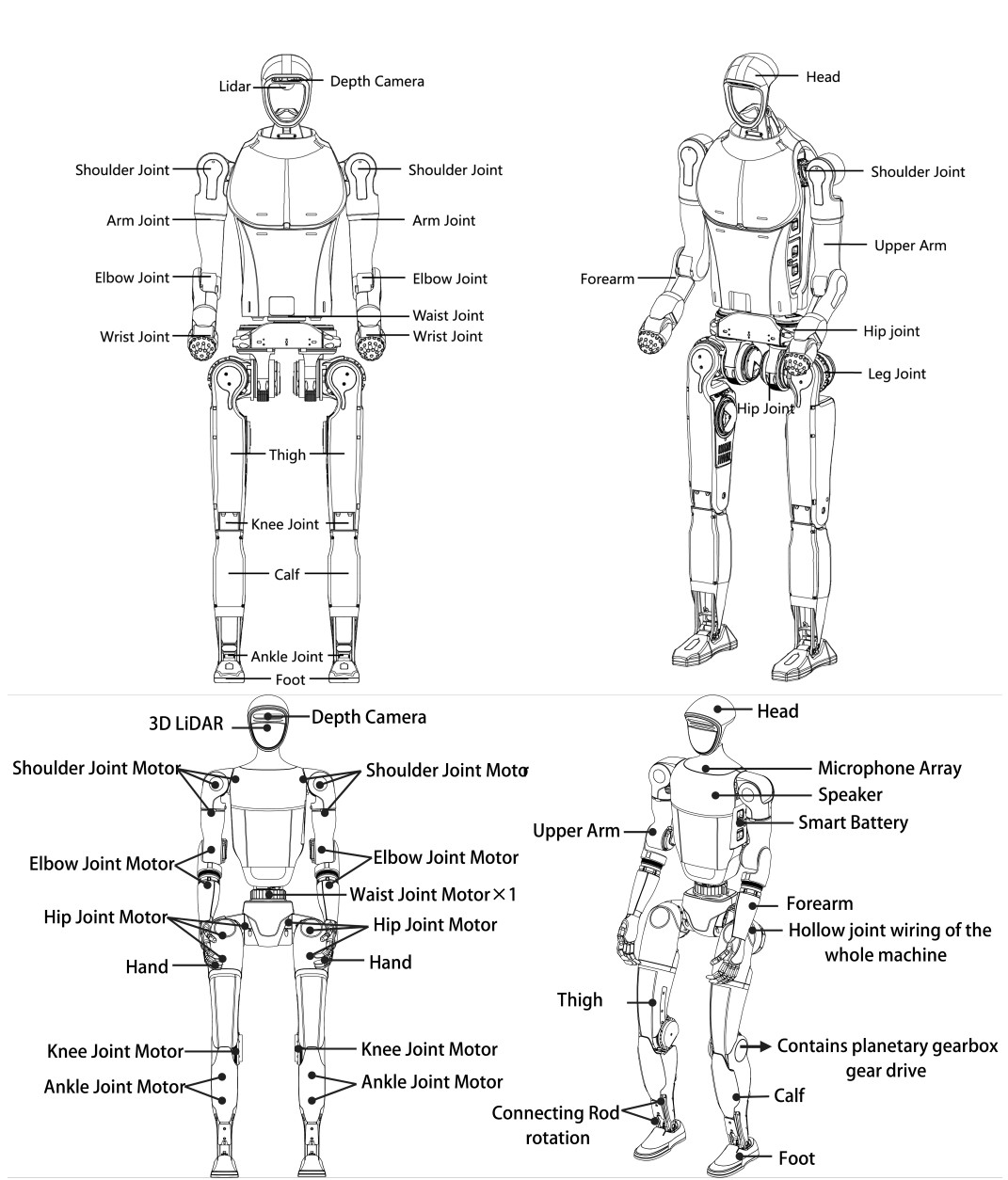

Figure 5: Joint names and positions on Unitree H1-2 and G1 robots

**Spatiotemporal synchronization** We use a single monotonic clock started at node initialization to time-stamp both the command loop and the sensor callback buffers. During acquisition, the node executes a fixed-rate control loop (50 Hz) and performs `rclpy.spin_once` with a short timeout each tick; the current monotonic time is appended to both `command_time_list` and `robot/joint_time_list`. This yields frame-accurate alignment between the actuation stream and the sensed state at the controller cadence. Since logging and control are co-located on the same machine, no cross-machine NTP is required; residual jitter is bounded by the loop period and handled in post-processing by resampling to a common time base when needed.

**Libraries** The implementation relies on `rclpy` (ROS 2), `numpy`, `torch` (policy inference/logging utilities), `mujoco` (simulation), `h5py` (file I/O), and `transforms3d` (frame utilities). All topics and message types (`LowState`, `LowCmd`, `MotorState`, `IMUState`) come from the `unitree_hg.msg` package.

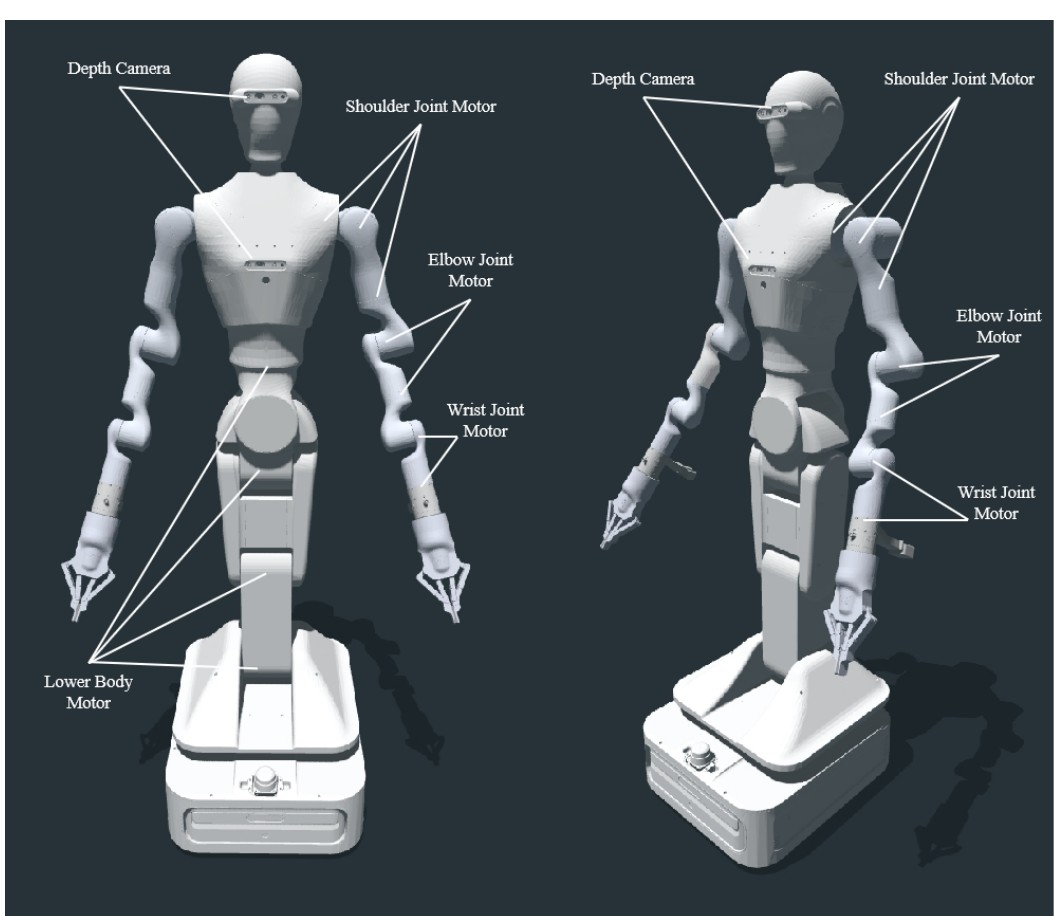

Figure 6: Joint names and positions on RealMan WR75S robot

### A.3.2 WHEELED HUMANOID ROBOT

We also collect motion execution data on dual-arm wheeled robots (RealMan). Our setup logs the full arm joint configurations along with actuator telemetry through UDP communication using the official RealMan API.

**Communication Setup** Data acquisition uses the RealMan official API with UDP communication. Position commands are sent to each arm at dedicated ports (8080, 8576), while real-time state data is received through UDP callbacks on separate ports (8089, 8090). The system registers

callback functions to process telemetry streams containing joint positions, velocities, currents, and temperatures.

**Data Recording Structure**    For each trial, we save timestamped datasets in HDF5 format with the following structure matching our dual-arm configuration:

- `command_time_list` (s): wall-clock timestamps when commands are issued.
- `command_val_list`: commanded action vectors for both arms concatenated (14-dimensional for dual 7-DoF arms).
- `robot1/joint_time_list`, `robot2/joint_time_list` (s): sensor timestamps for left and right arms respectively.
- `robot1/joint_angle_list`, `robot2/joint_angle_list`: joint positions in radians for each arm.
- `robot1/joint_velocity_list`, `robot2/joint_velocity_list`: joint velocities in rad/s for each arm.
- `robot1/joint_current_list`, `robot2/joint_current_list`: motor currents for each arm.
- `robot1/joint_temperature_list`, `robot2/joint_temperature_list`: actuator temperatures for each arm.
- `motion_name`, `slowdown_factor`, `current_time`: metadata for trial identification.

**Spatiotemporal synchronization**    We employ a unified monotonic clock initialized at data collection start to timestamp both command transmission and sensor reception. During execution, commands are sent via `rm_movej_canfd` API calls while the monotonic timestamp is recorded for both command and sensor streams. Since both command generation and sensor processing occur on the same machine with shared timing, cross-machine synchronization is unnecessary. The UDP callback mechanism ensures frame-accurate alignment between actuation commands and sensed states at the controller frequency. Residual timing jitter is bounded by the loop period and handled through post-processing resampling when temporal alignment is required for analysis. The system continuously monitors joint enable flags and error codes, with joint disable events prioritized as critical errors and other malfunctions classified as general errors, triggering immediate data cleanup and graceful termination.

### A.3.3    DATA SELECTION

We describe the amount of collected data in Section 3.2 and provide collection details in Section A.3. All data in these two sections are used as the training set. To evaluate the generalization ability of our operator, as stated in Section 5.1, we additionally collected an unseen-motion test set consisting of 100 sim–real pairs: 35 sequences at 0 kg, 23 at 1 kg, 22 at 2 kg, and 20 at 3 kg. The test set further spans three lower-body gaits in a 6:3:1 ratio for static stance, squat, and locomotion.

All motions used for collecting this test set are never used in the training dataset. To confirm the distinction between the two sets, we conduct t-SNE visualization and KS statistical testing (Figure 7). The results show that in the three motion-critical dimensions—dof_position, dof_velocity, and torque—the test dataset satisfies the zero-shot requirement described in our experiments.

### A.4    GAP ANALYSIS

### A.4.1    PD CONTROL

We use a basic joint-space proportional–derivative controller to track commanded trajectories with low latency. The proportional term corrects position error (stiffness), and the derivative term provides damping to reduce overshoot:

$$\tau \;=\; K_p \left(q_{\mathrm{cmd}} - q\right) \;+\; K_d \left(\dot{q}_{\mathrm{cmd}} - \dot{q}\right). \tag{11}$$

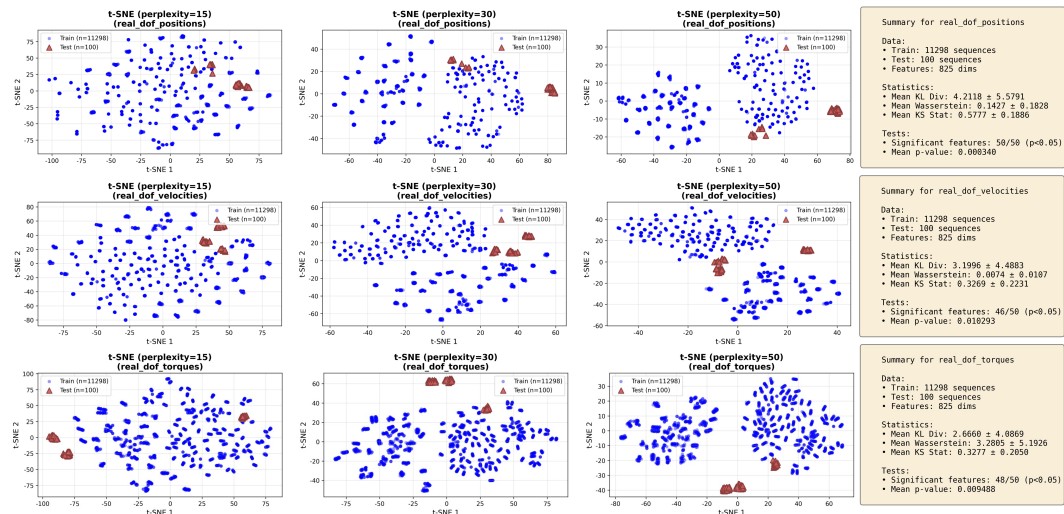

Figure 7: The t-SNE visualization and qualitative analysis results of the distribution of the train and the test dataset.

Here $q_{cmd}$ and $\dot{q}_{cmd}$ are the desired joint position/velocity, $q$ and $\dot{q}$ are the measured states, and $K_p, K_d$ (typically diagonal, positive) set tracking stiffness and damping. Optional gravity/feedforward terms can be added when needed, but the above is the minimal PD law.

In equation 1, $K_p(q_{cmd} - q) + K_d(\dot{q}_{cmd} - \dot{q})$ is the standard joint-space PD action (typically diagonal gains). The extra linear terms $K_v \dot{q}$ and $K_c \tanh(\dot{q}/\varepsilon)$ model viscous damping and smoothed Coulomb friction, respectively; $\varepsilon > 0$ regularizes the sign function to avoid chattering. The scalar (or diagonal) $P$ denotes the payload descriptor (e.g., mass/COM proxy). The bias $K_{payload} P$ provides a load-dependent offset, while $K_{P\sin} P \sin q$ and $K_{P\cos} P \cos q$ capture load-scaled gravity/-COM components in joint coordinates. Velocity/acceleration couplings $K_{P\dot{q}} P \dot{q}$ and $K_{P\ddot{q}} P \ddot{q}$ address payload-amplified damping/inertial effects. The constant $\tau_0$ compensates residual biases (e.g., calibration offsets).

Start from PD only ($K_p, K_d$), add $K_v, K_c$ to reduce overshoot and stick–slip, then introduce $K_{payload}, K_{P\sin}, K_{P\cos}$ for static/load gravity, and $K_{P\dot{q}}, K_{P\ddot{q}}$ for dynamic load effects; keep all gains bounded and $\varepsilon$ small enough to smooth $\tanh(\cdot)$ without degrading response.

### A.4.2 More analysis results

We present additional qualitative results here Figure 8 and Figure 9; further videos are provided in the supplementary materials.

### A.5 Nonlinear Operator

**What is an operator?** In contrast to learning a finite-dimensional mapping $f : \mathbb{R}^n \to \mathbb{R}^m$, operator learning targets a mapping between function spaces, $\mathcal{G} : \mathcal{U} \to \mathcal{V}$, where the input $u \in \mathcal{U}$ is itself a function and the output $\mathcal{G}(u) \in \mathcal{V}$ is another function. Practically, we observe $u$ via its sensor samples at locations $\{x_i\}_{i=1}^m$: $\{u(x_i)\}$, and we query the output at arbitrary $y$-locations to obtain values $\mathcal{G}(u)(y)$. This setup makes the learning objective function-to-function rather than pointwise regression, and enables generalization to unseen inputs $u$ and query points $y$.;

**Why not "learn a function" directly?** Classical approximation fits $(x, y)$ pairs for one target function. Operator learning instead aims to recover the rule that maps any admissible input function $u$ to an output function $\mathcal{G}(u)$. To make this learnable from data, we draw a diverse family of input functions—e.g., samples from Gaussian Random Fields (SE/RBF kernels with tunable length-scales/variances) and orthogonal polynomial expansions (e.g., Chebyshev with random coef-

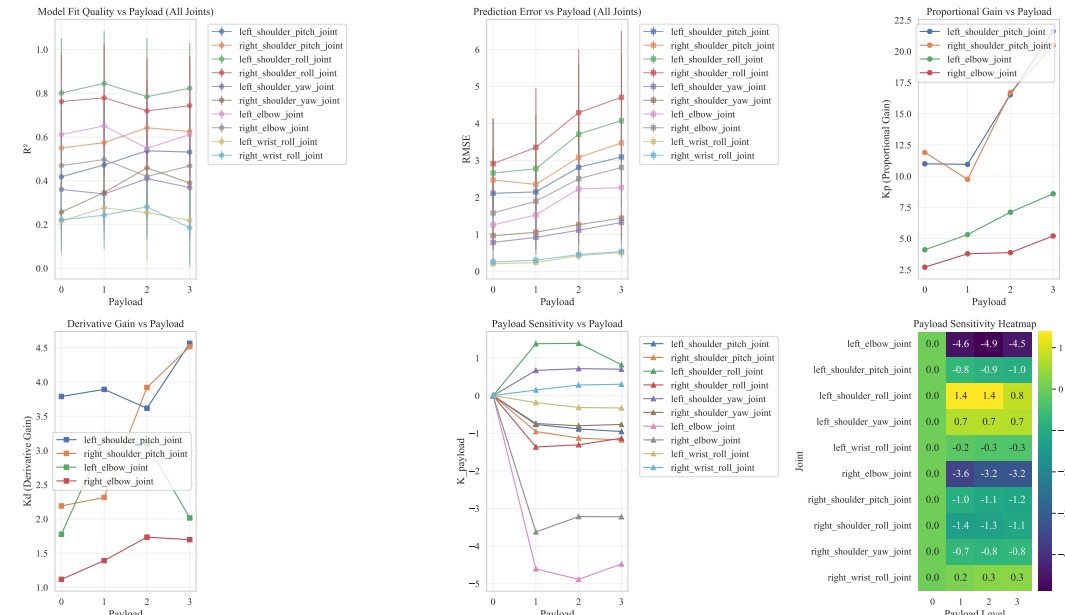

Figure 8: Data analysis on payload-related parameters

ficients)—so the model is trained across a rich subset of $\mathcal{U}$ rather than around a single curve. This ensures the learned mapping reflects an operator over a function class, not merely a single function fit.

**Low-rank/separable viewpoint** Many learned operators can be written (or approximated) in a separable, low-rank form

$$\widehat{\mathcal{G}}(u)(y) \;=\; \sum_{k=1}^{p} b_k(u)\, t_k(y), \tag{12}$$

where $b_k(u)$ are functionals of the input function (computed from its samples) and $t_k(y)$ are basis functions over the query variable $y$. This mirrors RKHS/separable-kernel and POD/SVD intuitions and clarifies the roles of "encode the input function" versus "encode the query location.";

We adopt this operator perspective to learn `GapONet`, a mapping from simulation context functions to hardware-space responses, so that the model predicts an output function of state/time given an input function describing simulated context—setting the stage for the DeepONet factorization introduced next.

### A.6 METHODS

#### A.6.1 WHY DO WE CHOOSE DEEPONET?

Our operator must (i) ingest simulation context functions with explicit payload conditioning, (ii) answer at arbitrary query points (current actions, payload) across heterogeneous robots and simulators, (iii) train under a closed-loop RL objective without requiring paired function-to-function supervision at every query, and (iv) support low-latency on-board inference.

We have considered some alternatives and trade-offs, for example:

- Fourier/Neural Operators (FNO family) (Li et al., 2020; Kovachki et al., 2023): excel on fixed grids with spectral convolutions, but rely on discretization tied to resolution/geometry; cross-morphology deployment (different joint layouts) typically needs regridding or retraining, and spectral blocks add latency on embedded hardware.

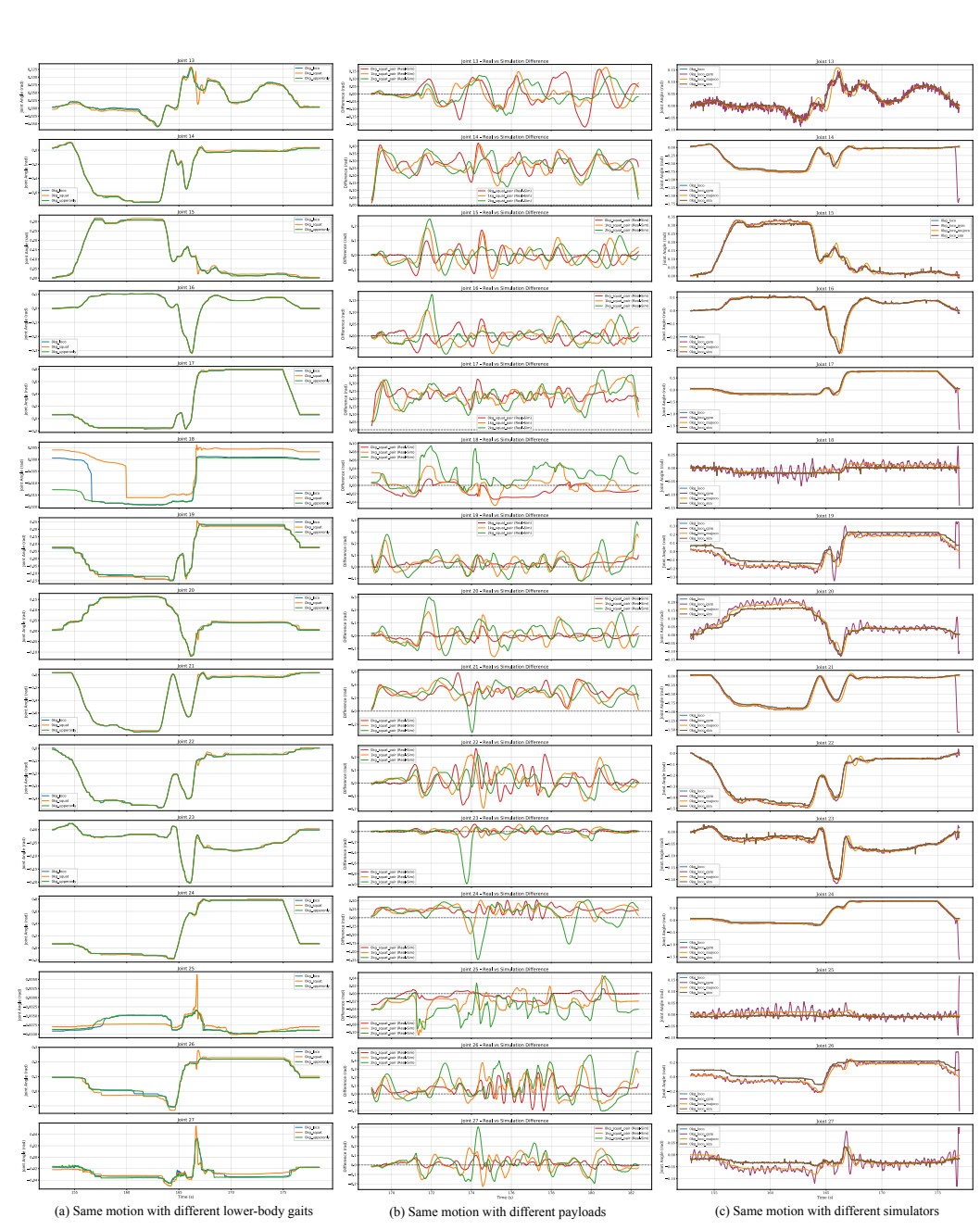

Figure 9: Results on all upper-body joints about the same motion with different payloads, simulations, and lower-body gaits.

- Graph/Galerkin/UNO-style operators (Kovachki et al., 2023): adapt to irregular meshes/graphs but require topology-aligned parameterization; when robots or sensor layouts change, weights/graphs must be remapped. Querying arbitrary state–time points is less natural than function–query separation. Capacity is high, but so are data and compute demands.

- Physics-informed neural operators (PINO): leverage known PDE residuals for sample efficiency, yet our residual field (sim→real actuation gap with delays/saturation) lacks a clean PDE form, making hard constraints difficult to specify and risking model-bias.

As for DeepONet's branch–trunk decomposition (Lu et al., 2019; 2021) aligns directly with our problem: the branch encodes context (multi-sensor histories, simulator traces, payload), and the trunk indexes continuous query variables (state/time/joint), producing residual action/torque values via a simple inner product. This yields (1) continuous space–time queries without grid lock-in, (2) clean conditioning on payload and robot-specific context without graph/topology rewiring, (3) RL-friendly training since supervision can be placed at arbitrary queried points along closed-loop rollouts, and (4) low-latency deployment because inference reduces to lightweight embeddings plus an inner product. Moreover, DeepONet comes with an operator-level universal approximation theorem that provides formal capacity guarantees for nonlinear operators (Lu et al., 2021), which we found attractive given the diversity of simulators, payloads, and hardware.

In summary, we choose DeepONet because its function–query factorization, theoretical operator approximation guarantees, and efficient, payload-conditioned querying match our requirements better than grid-bound spectral operators, topology-coupled graph variants, or physics-informed schemes that presume known PDE structure (Lu et al., 2019; 2021; Li et al., 2020; Kovachki et al., 2023). Our objective is to demonstrate that operator learning can achieve a mapping from simulation to reality, thereby aiding sim-to-real transfer. Determining the optimal operator architecture is outside the main scope of this work.

### A.6.2 THE DEFINITIONS OF SYMBOLS

- **Simulator $f^{\text{sim}}$.** We formalize simulators (e.g., Isaac Gym, MuJoCo) as functions $f^{\text{sim}} : S \times A \to S$ that compute the next state from the current state and an action. The state space $S$ typically includes joint parameters $(q, \dot{q})$, robot base states (e.g., root angular and linear velocities), and other environmental variables. In our framework, we decompose a state $s \in S$ based on its influence on joint actuation: $s^{\xi}$ denotes the states that directly influence the actuation of the joints, $p$ represents payload, and $s^{\text{other}}$ encompasses all remaining states that do not affect joint actuation. Consequently, the simulator can be expressed as $f^{\text{sim}}(s, a) = f^{\text{sim}}(s^{\text{other}}, s^{\xi}, p, a)$. To focus on the joints, we define the desired state transition as $\Delta f^{\text{sim}}(s^{\xi}, x) = (f^{\text{sim}}(s^{\text{other}}, s^{\xi}, p, a_x))_j - s^{\xi}_j$, where the subscript $j$ extracts only the joint-related states (position and velocity) for transition computation, excluding uninfluential states such as root velocities.

- **Clarification on States $s^t$ and $s^{\xi}$.** The description in Section 4 primarily uses $s^{\xi}$ to denote states, irrespective of the domain (simulation or real). However, specific equations (e.g., Equation (9)) employ $s^t$ to emphasize that the state belongs to a trajectory at a specific time $t$ in **TWINS**. Each trajectory forms a dynamic path $\xi_t$, and thus $s^t$ corresponds precisely to $s^{\xi_t}$.

- **Actuation Functions $U^{\text{sim}}_{\xi}$, $U^{\text{real}}_{\xi}$ and $U^{\text{surr}}_h$.** The actuation function $U^{\text{sim}}_{\xi}$ is defined as $\Delta f^{\text{sim}}(s^{\xi}_{\text{sim}}, \cdot)$ following Section 4.1. Its output $U^{\text{sim}}_{\xi}(a, p) = \Delta f^{\text{sim}}(s^{\xi}_{\text{sim}}, (a, p))$ represents the concatenation of delta joint position and delta joint velocity in $\mathbb{R}^{2J}$, where $J$ is the total number of joints. The real actuation function $U^{\text{real}}_{\xi}$ shares a similar formulation and output dimension, but is defined using the real-world dynamics $f^{\text{real}}$ in place of $f^{\text{sim}}$. In contrast, the surrogate actuation function $U^{\text{surr}}_h$ also outputs values in $\mathbb{R}^{2J}$, but differs in representation: it is the output of a sensor predictor, implemented as a neural network, with inputs from $h$-step joint position, velocity and action history.

- **Sensor Values, Branch Net $\mathcal{B}$ and Trunk Net $\mathcal{T}$.** Sensor values represent the state transitions of joints under a specific dynamics parameter $\xi$. The concatenated sensor vector

$S(U_\xi)$ lies in $\mathbb{R}^{2kJ}$, where $2J$ corresponds to the position and velocity changes across $J$ joints, and $k$ denotes the number of sensor locations. For details on the computation of $S$, refer to Section 4.3. The Branch Net $\mathcal{B}$ and Trunk Net $\mathcal{T}$ are both implemented as standard multi-layer perceptrons (MLPs); their specific configurations are provided in Section A.8.

- **Operator, $\mathcal{G}$ and $G_\theta$.** $G_\theta$ is an intermediate representation formed by the element-wise dot-product of the Brand Net $\mathcal{B}$ and the Trunk Net $\mathcal{T}$, where $\theta$ denotes the combined parameters of both $\mathcal{B}$ and $\mathcal{T}$. If the dynamics parameter $\xi$ can be represented as a real-number vector, then for any input $y$, $G_\theta$ is deterministic, differentiable, and amenable to direct optimization. However, rather than supervising $G_\theta$ directly, we interpret its outputs not as state transitions, but as delta actions. We subsequently introduce $\Delta f^{\text{sim}}$ to formulate $\mathcal{G}$ as the final operator. This design choice is intrinsically linked to our decision to use Reinforcement Learning (RL) in place of supervised learning; see Section A.6.3 for further justification.

### A.6.3 WHY DO WE CHOOSE REINFORCEMENT LEARNING

**Computational Prohibitivity.** Direct computation of sensor values for each $\xi$ is computationally prohibitive under our setting, which requires evaluating the actuation function $U_\xi$ at $k$ fixed locations $\{x_i\}_{i=1}^k$. To illustrate, consider a continuous motion execution involving a fixed trajectory of $x$ and $\xi$ correlated with the current motion playback time. Direct evaluation of sensor values would require saving a simulation checkpoint at every timestep $t$, executing all $\{x_i\}_{i=1}^k$ in simulation, and retrieving the corresponding values. Subsequently, all parallel environments would need to be reset to $\xi(t)$ before proceeding with the execution of $x$ from the motion incorporating corrections from our operator. This process significantly impedes execution efficiency: computing $k$ sensor values would slow down the motion trajectory execution by at least a factor of $1/k$. To mitigate this, we introduce a sensor predictor, thereby constructing a surrogate actuation function space.

**Non-Differentiable Simulators.** Once the surrogate actuation function space is constructed, the remaining challenge is to optimize the operator that minimizes the multi-step transition discrepancy between simulation and the real robot. However, this optimization objective depends on the simulator's internal dynamics—contact events, actuator nonlinearities, sensor latency, and frictional discontinuities—which are inherently *non-differentiable*. As a result, a supervised-learning formulation would require backpropagating through the simulator, which is infeasible under GPU-based physics engines such as Isaac Gym/Isaac Sim. In contrast, reinforcement learning treats the simulator as a black-box transition model and optimizes the operator purely from trajectory-level rewards, without requiring differentiability. This makes RL the only practical and efficient optimization framework for training our operator in the presence of non-smooth, non-differentiable sim-to-real dynamics.

### A.7 EXPERIMENT

#### A.7.1 METRICS

We report two metric families: (i) gap distribution (Table 1: large-gap ratio(LGR), interquartile range (IQR), and gap range) and (ii) kinematic quality of lower-body (Table 2: smoothness, trajectory consistency, and robustness). All metrics are computed per run and then aggregated by payload mass (the environment groups trials by mass buckets).

Let $q_t^{\text{real}}, q_t^{\text{sim}}$ be joint trajectories (or end-effector signals) sampled at uniform $\Delta t$. Define the gap $g_t = q_t^{\text{real}} - q_t^{\text{sim}}$ and its absolute value $|g_t|$. Central-difference operators approximate derivatives.

**Large-gap ratio (Table 1)** Fraction of samples with absolute joint error exceeding a threshold (0.5 rad by default):

$$\text{Large-gap ratio} = \frac{\big|\{(t,i) : |g_{t,i}| \geq \tau\}\big|}{\big|\{(t,i)\}\big|}, \qquad \tau = 0.5 \text{ rad.} \tag{13}$$

Captures the frequency of serious deviations.

We adopt the commonly used 0.5 rad threshold, which prior work Zhang et al. (2022); Sun (2023) employs as a perturbation magnitude for identifying severe tracking failures rather than normal

---

**Algorithm 1: `GapONet` Training with PPO in Simulation**

---

**Input:** Simulator $f^{\text{sim}}$, real-world dataset $\mathcal{D}$, learning rates $\alpha_\theta, \alpha_\phi$, parallel environment count $B$, PPO parameters $L_{\text{buffer}}, \gamma, \lambda$, operator training steps $N$, sensor model training steps $N_{\text{sensor}}$, history length $N_h$

**Initialize:** Network parameters $\theta$ for $G_\theta$, $\phi$ for $S_\phi$, PPO value function $V_\psi$, PPO buffer $\mathcal{D}_{\text{PPO}}$

```
// Sensor Model Pre-training Phase
```
**for** *iteration* $\leftarrow 1$ **to** $N_{sensor}$ **do**
     Sample initial states $s_0 \in \mathbb{R}^{B \times 2J}$;            `// Joint positions and velocities`
     Sample task parameters $p \in \mathbb{R}^P$;            `// P = 1 for payload in our settings`
     Sample action sequence $\{a_t\}_{t=0}^{h-1}$ where $a_t \in \mathbb{R}^{B \times J}$
     `// Rollout in simulator to collect dynamics data`
     **for** $t \leftarrow 0$ **to** $h-1$ **do**
         $s_{t+1} \leftarrow f^{\text{sim}}(s_t, a_t, p)$;            `// State transition in simulation`
     **end**
     `// Compute sensor model training targets`
     $I \leftarrow \{(s_t, a_t) \mid t = 0, \dots, h-1\}$;            `// History input`
     $\mathcal{L}_\phi \leftarrow \text{MSE}(s_h - s_{h-1}, S_\phi(I))$;            `// Predict state transitions`
     $\phi \leftarrow \phi - \alpha_\phi \nabla_\phi \mathcal{L}_\phi$;            `// Update sensor model`
**end**
```
// Operator Learning Phase with PPO
```
Initialize all environments as done
**for** *iteration* $\leftarrow 1$ **to** $N$ **do**
     **foreach** *environment marked done* **do**
         Sample trajectory from $\mathcal{D}$ with $p \in \mathbb{R}^P$, $\{a_t\} \in \mathbb{R}^{B \times J}$, $\{s_{\text{real}}^t\} \in \mathbb{R}^{B \times S}$
         Reset environment to initial state $s_{\text{real}}^0$
     **end**
     `// Compute operator inputs and corrections`
     Construct history input $I$ from recent states and actions
     Compute surrogate sensor values: $\hat{S} \leftarrow S_\phi(I)$
     Form query vector: $y \leftarrow (a_t, p)$
     Compute action correction: $\Delta a_t \leftarrow G_\theta(h, y)$;            `// Using Equation (5)`
     `// Step simulator with corrected actions`
     $\Delta \hat{s}_{\text{sim}} \leftarrow \Delta f^{\text{sim}}(s_{\text{sim}}^t, a_t + \Delta a_t, p)$
     Compute reward: $r_t \leftarrow -w \left\| (s_{\text{real}}^{t+1} - s_{\text{real}}^t) - \Delta \hat{s}_{\text{sim}} \right\|_2^2$;            `// Using Equation (9)`
     `// Store experience for PPO`
     Add transition $(s_{\text{sim}}^t, \Delta a_t, r_t, s_{\text{sim}}^{t+1})$ to $\mathcal{D}_{\text{PPO}}$
     **if** *iteration* $\mod L_{buffer} = 0$ **then**
         $\theta, \psi \leftarrow \text{PPO\_Update}(\mathcal{D}_{\text{PPO}}, \gamma, \lambda, \alpha_\theta)$;            `// Update policy and value networks`
         Clear buffer: $\mathcal{D}_{\text{PPO}} \leftarrow \emptyset$
     **end**
**end**
**Output:** Trained parameters $\theta^*, \phi^*, \psi^*$

---

fluctuations. This value is intentionally set far above typical joint-tracking errors in robot control. Rrrors exceeding 0.5 rad correspond to catastrophic sim-to-real failures, making LGR a meaningful indicator of such gaps.

To verify that the 0.5 rad threshold meaningfully reflects the natural distribution of the sim–real gap, we analyzed the entire dataset Figure 10. The histogram shows that while most gaps are small, there is a clear heavy tail, indicating that large deviations do occur and should be detected by a threshold-based metric. The CDF curve further confirms that about 20% of all samples lie above 0.5 rad, meaning the threshold captures a substantial portion of true large-error events rather than rare outliers. The percentile plot shows that 0.5 rad lies between the 75th and 90th percentiles, aligning with the onset of severe deviations. Finally, the per-payload density curves demonstrate that this heavy tail persists across payloads, so 0.5 rad consistently separates normal fluctuations from genuinely large tracking failures. Together, these results show that the 0.5 rad threshold is not arbitrary but well matched to the intrinsic structure of the gap distribution.

**Gap IQR (Table 1)** Dispersion of absolute errors via the interquartile range:

$$\mathcal{G} = \big\{ |g_{t,i}| \ : \ t = 1, \dots, T, \ i = 1, \dots, J \big\}, \qquad \text{IQR} = Q_{0.75}(\mathcal{G}) - Q_{0.25}(\mathcal{G}). \tag{14}$$

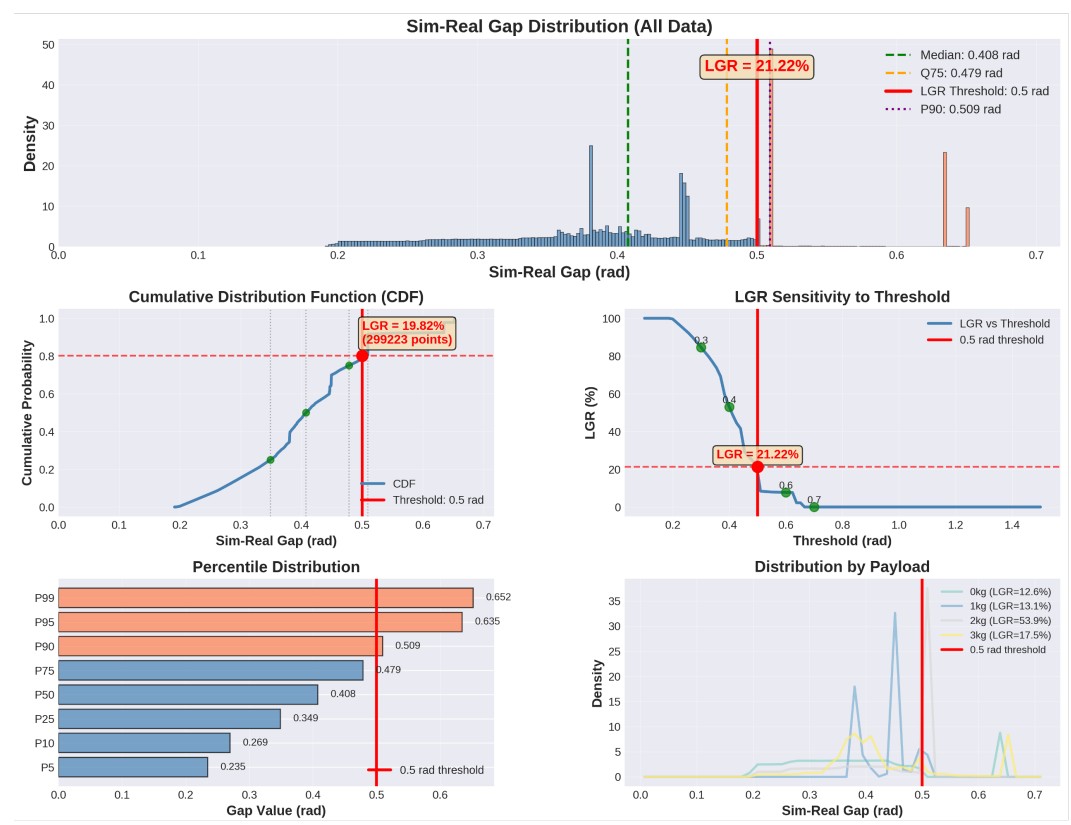

Figure 10: Sim-real gap distribution analysis.

Lower is a tighter error distribution.

**Gap range (Table 1)** Extreme-case spread of absolute errors:

$$\text{Range} = \max(|g|) - \min(|g|). \tag{15}$$

Highlights worst-case variability.

**Trajectory consistency (Table 2)** Discrepancy in the *rate-of-change of velocity* (a curvature-like signal) between real and simulated motion:

$$v\_t^{\text{real}} = \nabla q\_t^{\text{real}}, \quad v\_t^{\text{sim}} = \nabla q\_t^{\text{sim}}, \quad \kappa\_t^{\text{real}} = \nabla v\_t^{\text{real}}, \quad \kappa\_t^{\text{sim}} = \nabla v\_t^{\text{sim}}, \tag{16}$$

$$\text{TrajectoryConsistency} = \frac{1}{T} \sum_{t=1}^{T} \left| \kappa_t^{\text{real}} - \kappa_t^{\text{sim}} \right|. \tag{17}$$

Smaller values indicate that the simulator reproduces the evolution of motion patterns more faithfully.

**Smoothness (Table 2)** Discrepancy in *accelerations* between real and simulated trajectories:

$$\text{Smoothness} = \frac{1}{T} \sum_{t=1}^{T} \left| a_t^{\text{real}} - a_t^{\text{sim}} \right|, \qquad a_t^{\text{real}} = \nabla^2 q_t^{\text{real}}, \ a_t^{\text{sim}} = \nabla^2 q_t^{\text{sim}}. \tag{18}$$

Lower scores mean closer kinematic smoothness to real motion.

**Robustness (Table 2)**   Sensitivity of the sim–real gap to *measurement noise*. For noise levels $\sigma \in \{\sigma_1, \ldots, \sigma_K\}$,

$$\text{Robustness} = \frac{1}{K} \sum_{k=1}^{K} \left[ \frac{1}{T} \sum_{t=1}^{T} \left| \left( q_t^{\text{real}} + \epsilon_t^{(k)} \right) - \left( q_t^{\text{sim}} + \tilde{\epsilon}_t^{(k)} \right) - g_t \right| \right], \tag{19}$$

$$g_t = q_t^{\text{real}} - q_t^{\text{sim}}, \quad \epsilon_t^{(k)}, \tilde{\epsilon}_t^{(k)} \sim \mathcal{N}(0, \sigma_k^2). \tag{20}$$

Smaller values indicate that the evaluation is stable under realistic perturbations.

Each motion is run at least six times. For each run, we compute every metric (optionally per joint and then averaged); otherwise, only real-stream statistics are used as specified by each metric. We then aggregate runs by payload/mass buckets and report means with standard errors. All three metrics are discrepancy-style measures; by construction, **smaller values indicate better performance**.

### A.7.2   LOCOMOTION TRAJECTORY TRACKING

We generate locomotion commands using a phase-based trajectory: a normalized phase $\phi \in [0, 1)$ advances at the control rate and indexes a trapezoidal base-velocity profile (accelerate–cruise–decelerate–pause). Forward and backward segments alternate automatically, while lateral velocity and yaw rate remain zero unless specified. The phase schedules lower-body gait timing and yields desired joint trajectories for the legs, tracked by a joint-space PD controller at 50 Hz with torque/rate limits and safety checks.

Fixed start pose and heading. Each real-robot run starts from the same world-frame pose—a fixed position and heading—followed by a short smooth interpolation into the nominal stand pose before the phase route is enabled. This ensures repeatable initial conditions, so the resulting base trajectory in SE(2) (odometry or motion-capture) can be compared across runs to assess tracking quality, drift, and sim–real alignment. Commands and sensor streams share a monotonic timestamp, keeping phase, velocity setpoints, and measured joint/IMU signals time-aligned for evaluation.

### A.7.3   ABLATION ON OPERATOR VS. MLP

We provide an ablation study comparing the proposed `GapONet` architecture against **a standard high-capacity MLP** that is likewise conditioned on the payload and the simulation context, however, its architecture differs fundamentally from the MLP baseline used in Section A.8.3. Specifically, we replace the branch–trunk networks with a single MLP placed after the sensor model, which we refer to as **MLP-Sensor**. Since our sensor model contains explicit history information, we additionally compare against two alternative baselines: (i) an MLP that directly receives the raw history without any processing (**MLP-History**), and (ii) a minimal MLP that does not incorporate any history information (**MLP-Pointwise**).

To further validate our conclusions, we also construct MLP variants with different parameter scales—**Small** ([256, 128, 128]), **Medium** ([512, 256, 128]), and **Large** ([512, 512, 512])—and demonstrate that merely increasing model capacity does not yield improved performance; rather, architectural design is essential. In total, this yields nine additional baselines: **MLP-Pointwise-Small/Medium/Large, MLP-History-Small/Medium/Large, and MLP-Sensor-Small/Medium/Large**.

As shown in Table 3, by comparing **MLP-Sensor-Small/Medium/Large** with **MLP-History-Small/Medium/Large**, we observe that when both models receive history information, the sensor predictor provides limited benefit for the LGR and IQR metrics, but leads to a substantial improvement in the Range metric. However, both variants remain noticeably inferior to `GapONet`, indicating that the zero-shot generalization capability primarily arises from the operator-learning formulation rather than from the residual network structure itself.

By comparing **MLP-Pointwise-Small/Medium/Large** with `GapONet`, we find that their zero-shot performance differs substantially. Although the MLP-Pointwise variants can achieve LGR scores close to `GapONet` in the 0kg setting, the gap widens consistently as the payload increases: both

the magnitude and frequency of the errors grow significantly. This directly demonstrates that learning operators of actuator functions is necessary and superior to pointwise mappings, and that the insufficiency of pointwise modeling fundamentally limits its ability to generalize.

In addition to the zero-shot comparisons above, we observe distinct training behaviors across the three architectures and model sizes. As shown in Figure 11, under a unified network capacity, the sensor-based architecture achieves the lowest joint angle error during training, followed by the pointwise model, while the history-augmented MLP exhibits the highest error. When using the pointwise method exclusively, training error increases with model size. The result suggests that the sensor model effectively captures simulator dynamics and facilitates learning. This also confirms that the poor performance of MLPs **is not due to insufficient capacity**. Furthermore, even though the MLP-Sensor achieves training errors nearly as low as `GapONet`, it still underperforms on the test set, indicating its **limited generalization ability.**

Table 3: Ablation study comparing `GapONet` with different MLP architectures of matched capacity.

| Method | 0 kg | | | 1 kg | | |
|---|---|---|---|---|---|---|
| | LGR(%) (↓) | IQR (↓) | Range (↓) | LGR(%) (↓) | IQR (↓) | Range (↓) |
| MLP-Pointwise-Small | $\mathbf{0.08}^{\pm 0.04}$ | $0.093^{\pm 0.016}$ | $0.646^{\pm 0.083}$ | $0.77^{\pm 0.80}$ | $0.213^{\pm 0.009}$ | $0.670^{\pm 0.061}$ |
| MLP-Pointwise-Medium | $0.08^{\pm 0.05}$ | $0.095^{\pm 0.010}$ | $0.651^{\pm 0.087}$ | $0.71^{\pm 0.79}$ | $0.214^{\pm 0.010}$ | $0.665^{\pm 0.06}$ |
| MLP-Pointwise-Large | $0.08^{\pm 0.05}$ | $0.097^{\pm 0.008}$ | $0.653^{\pm 0.090}$ | $0.76^{\pm 0.88}$ | $0.206^{\pm 0.012}$ | $0.665^{\pm 0.059}$ |
| MLP-History-Small | $0.10^{\pm 0.06}$ | $0.098^{\pm 0.009}$ | $0.662^{\pm 0.087}$ | $1.04^{\pm 1.10}$ | $0.213^{\pm 0.009}$ | $0.679^{\pm 0.074}$ |
| MLP-History-Medium | $0.09^{\pm 0.06}$ | $0.096^{\pm 0.009}$ | $0.658^{\pm 0.077}$ | $0.98^{\pm 1.06}$ | $0.213^{\pm 0.008}$ | $0.682^{\pm 0.069}$ |
| MLP-History-Large | $0.11^{\pm 0.05}$ | $0.111^{\pm 0.007}$ | $0.675^{\pm 0.098}$ | $1.17^{\pm 1.26}$ | $0.197^{\pm 0.010}$ | $0.674^{\pm 0.058}$ |
| MLP-Sensor-Small | $0.10^{\pm 0.06}$ | $0.097^{\pm 0.010}$ | $0.667^{\pm 0.081}$ | $1.05^{\pm 1.32}$ | $0.125^{\pm 0.009}$ | $0.578^{\pm 0.065}$ |
| MLP-Sensor-Medium | $0.09^{\pm 0.06}$ | $0.094^{\pm 0.009}$ | $0.658^{\pm 0.081}$ | $0.90^{\pm 1.10}$ | $0.123^{\pm 0.008}$ | $0.572^{\pm 0.063}$ |
| MLP-Sensor-Large | $0.09^{\pm 0.05}$ | $0.093^{\pm 0.009}$ | $0.651^{\pm 0.076}$ | $0.87^{\pm 1.03}$ | $0.128^{\pm 0.007}$ | $0.572^{\pm 0.066}$ |
| `GapONet` (Ours) | $\mathbf{0.09}^{\pm 0.03}$ | $\mathbf{0.093}^{\pm 0.016}$ | $\mathbf{0.449}^{\pm 0.117}$ | $\mathbf{0.22}^{\pm 0.11}$ | $\mathbf{0.115}^{\pm 0.013}$ | $\mathbf{0.537}^{\pm 0.148}$ |

| Method | 2 kg | | | 3 kg | | |
|---|---|---|---|---|---|---|
| | LGR(%) (↓) | IQR (↓) | Range (↓) | LGR(%) (↓) | IQR (↓) | Range (↓) |
| MLP-Pointwise-Small | $2.34^{\pm 1.27}$ | $0.204^{\pm 0.011}$ | $0.775^{\pm 0.069}$ | $11.19^{\pm 1.50}$ | $0.354^{\pm 0.011}$ | $0.969^{\pm 0.093}$ |
| MLP-Pointwise-Medium | $2.26^{\pm 1.21}$ | $0.204^{\pm 0.009}$ | $0.774^{\pm 0.076}$ | $11.06^{\pm 1.32}$ | $0.352^{\pm 0.011}$ | $0.964^{\pm 0.102}$ |
| MLP-Pointwise-Large | $2.19^{\pm 1.23}$ | $0.200^{\pm 0.011}$ | $0.780^{\pm 0.075}$ | $10.76^{\pm 1.53}$ | $0.355^{\pm 0.011}$ | $0.976^{\pm 0.096}$ |
| MLP-History-Small | $2.45^{\pm 1.10}$ | $0.204^{\pm 0.009}$ | $0.794^{\pm 0.066}$ | $11.36^{\pm 1.39}$ | $0.356^{\pm 0.011}$ | $1.00^{\pm 0.118}$ |
| MLP-History-Medium | $2.53^{\pm 1.10}$ | $0.205^{\pm 0.009}$ | $0.799^{\pm 0.070}$ | $11.44^{\pm 1.42}$ | $0.356^{\pm 0.010}$ | $1.00^{\pm 0.112}$ |
| MLP-History-Large | $2.43^{\pm 1.16}$ | $0.199^{\pm 0.009}$ | $0.792^{\pm 0.077}$ | $10.74^{\pm 1.57}$ | $0.358^{\pm 0.011}$ | $0.999^{\pm 0.104}$ |
| MLP-Sensor-Small | $2.64^{\pm 1.32}$ | $0.207^{\pm 0.010}$ | $0.618^{\pm 0.065}$ | $11.62^{\pm 1.40}$ | $0.357^{\pm 0.012}$ | $0.991^{\pm 0.137}$ |
| MLP-Sensor-Medium | $2.59^{\pm 1.14}$ | $0.207^{\pm 0.009}$ | $0.591^{\pm 0.079}$ | $11.79^{\pm 1.31}$ | $0.357^{\pm 0.010}$ | $0.992^{\pm 0.124}$ |
| MLP-Sensor-Large | $2.66^{\pm 1.27}$ | $0.208^{\pm 0.009}$ | $0.607^{\pm 0.067}$ | $12.05^{\pm 1.25}$ | $0.458^{\pm 0.010}$ | $0.995^{\pm 0.114}$ |
| `GapONet` (Ours) | $\mathbf{0.39}^{\pm 0.10}$ | $\mathbf{0.161}^{\pm 0.004}$ | $\mathbf{0.578}^{\pm 0.112}$ | $\mathbf{0.84}^{\pm 0.23}$ | $\mathbf{0.317}^{\pm 0.005}$ | $\mathbf{0.498}^{\pm 0.157}$ |

### A.7.4 SUPPLEMENTARY EXPERIMENTS DURING THE REBUTTAL PERIOD

**Compare to domain randomization.** At present, Domain Randomization (DR) is indeed one of the most widely used strategies for addressing sim-to-real transfer. However, DR and `GapONet` differ in a fundamental way. Our operator learns a structured residual model that captures and corrects the dynamical discrepancies between simulation and the real world, producing a delta action that improves the execution of a given command on the physical robot. In contrast, DR expands the parameter distribution of the simulator during training to improve robustness, and directly outputs the next action at each timestep. While effective for robustness, DR does not explicitly model nor correct the structural components of the sim-to-real gap. From a modeling perspective, DR seeks to

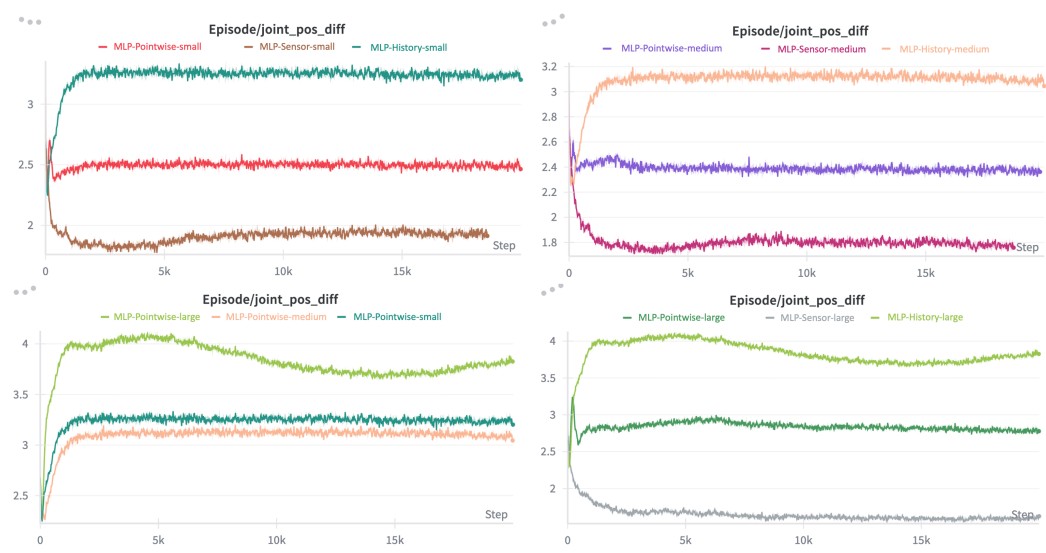

Figure 11: Joint position difference curves during training.

produce a more robust action, whereas **GapONet** produces a residual term that makes that action actually work on real hardware.

Building on this distinction, we constructed a more direct and quantitative comparison against a strong DR baseline for both experiments—*Zero-shot Motion Tracking* and *Locomotion Trajectory Tracking*. For Experiment 1, we trained a whole-body tracker with domain randomization over payload mass (0,1,2,3kg). The model receives the same inputs as **GapONet** ($(q_t, \dot{q}_t)$) and outputs the next-step action ($a_{t+1}$). The results have been added to Table 1, along with corresponding videos in the supplementary material. The results show that DR-only control struggles to reach the target joint angles for zero-shot motions, and as payload increases, the robot becomes increasingly unstable. With a 1,2,3 kg payload, the DR policy fails to execute the motion entirely.

For Experiment 2, the lower-body controller provided in the main paper already uses DR to ensure stable locomotion when the upper body is fixed. In the new ablation, we provide videos of the same controller **without** domain randomization (included in the supplementary material). As shown, the robot exhibits continuous swaying even during standing, and cannot serve as a valid comparison baseline. Table 4

**Compare to nonlinear system identification.**    As described in Section A.8.3, our nonlinear system identification follows standard practice, fitting rigid-body dynamics using both MLP- and SVR-based estimators. The *Zero-shot Motion Tracking* results in Table 1 show that all three nonlinear SysID methods yield similar performance, with the kernel-based estimator achieving slightly lower Large Gap Ratio and Gap Range. In contrast, the network-based estimator shows weaker zero-shot generalization, indicating that overfitting to the training dataset cannot compensate for unseen motions. As for *Locomotion Trajectory Tracking* Table 2, all three SysID variants perform comparably across all metrics, suggesting that nonlinear SysID alone neither improves nor degrades locomotion tracking performance in this setting.

### A.7.5   COMPUTATIONAL OVERHEAD

Table 4: Sim-to-real gap in locomotion trajectory tracking on an unseen humanoid robot.

| Method | Trajectory Consistency ($\downarrow$) | Smoothness ($\downarrow$) | Robustness ($\downarrow$) |
|---|---|---|---|
| PD control | $20.33^{\pm 1.982}$ | $53.76^{\pm 0.257}$ | $10.16^{\pm 0.007}$ |
| PD control w/o DR | - | - | - |
| MLP | $19.18^{\pm 0.919}$ | $53.48^{\pm 0.343}$ | $10.15^{\pm 0.027}$ |
| Transformer | $19.13^{\pm 0.689}$ | $53.57^{\pm 0.290}$ | $10.14^{\pm 0.007}$ |
| System Identification | $19.16^{\pm 0.489}$ | $\mathbf{24.99}^{\pm 0.298}$ | $10.14^{\pm 0.011}$ |
| Network-based SysID | $19.05^{\pm 0.175}$ | $25.47^{\pm 0.408}$ | $10.15^{\pm 0.157}$ |
| Kernel-based SysID | $19.11^{\pm 0.465}$ | $25.84^{\pm 0.246}$ | $10.17^{\pm 0.078}$ |
| **GapONet** (Ours) | $\mathbf{18.78}^{\pm 1.147}$ | $53.36^{\pm 0.486}$ | $\mathbf{10.13}^{\pm 0.167}$ |

Table 5: Real-time inference cost of each method on real robot

| | MLP | Transformer | **GapONet** |
|---|---|---|---|
| Time(s) | 0.0001600 | 0.0001181 | 0.0003764 |

### A.7.6 EXPLANATION OF THE TWO EXPERIMENT SETTINGS

This section provides a mathematically coherent justification for the residual-action design used in both of our experiments. We formalize why the operator output $G_\theta(\xi, y)$ is **added** to the simulator command during training, but **subtracted** from the real hardware command during online deployment. For any query $y \in A \times P$, the simulated and real actuation functions yield

$$U_\xi^{\text{sim}}(y) = \Delta f^{\text{sim}}(s_{\text{sim}}^\xi, y), \qquad U_\xi^{\text{real}}(y) = \Delta f^{\text{real}}(s_{\text{real}}^\xi, y), \tag{21}$$

where the $\Delta f^{\text{real}}$ can be treated as the real robot excution process. And their discrepancy is

$$\delta_\xi(y) = U_\xi^{\text{real}}(y) - U_\xi^{\text{sim}}(y). \tag{22}$$

**Residual addition in simulation** During training, the operator output $G_\theta(\xi, y_t)$ produces a corrective delta action added to the simulator command:

$$\mathcal{G}(U_\xi^{\text{sim}})(y_t) = \Delta f^{\text{sim}}\left(s_{\text{sim}}^\xi, \, a_t + G_\theta(\xi, y_t)\right), \tag{23}$$

where $y_t = (a_t, p)$. Linearizing the simulator dynamics around $a_t$ gives $\Delta f^{\text{sim}}(s_{\text{sim}}^\xi, a_t + G_\theta) \approx \Delta f^{\text{sim}}(s_{\text{sim}}^\xi, a_t) + J_{a_t}^{\text{sim}} G_\theta(\xi, y_t)$, where $J_{a_t}^{\text{sim}}$ is the simulator's action "Jacobian", defined as

$$J_a^{\text{sim}} = \left(\frac{\Delta s_i}{\Delta a_j}\right)_{ij}, \tag{24}$$

where $\frac{\Delta s_i}{\Delta a_j}$ represents the relative difference of desired state to action under $\Delta t$ of simulation . To match the real transition, i.e., $\Delta f^{\text{sim}}(s_{\text{sim}}^\xi, a_t + G_\theta) \approx \Delta f^{\text{real}}(s_{\text{real}}^t, a_t)$. The correction must satisfy

$$J_{a_t}^{\text{sim}} G_\theta(\xi, y_t) \approx \delta_\xi(y_t), \tag{25}$$

showing that the operator learns the action-space residual necessary to *inject missing real-world dynamics into the simulator*.

**Residual subtraction in the real world** On hardware, the goal is inverted: we seek a corrected real command $a_t'$ such that the real dynamics match the simulator's nominal prediction: $\Delta f^{\text{real}}(s_{\text{real}}^\xi, a_t') \approx \Delta f^{\text{sim}}(s_{\text{sim}}^\xi, a_t^{\text{real}})$. Leveraging Equation (23) gives $\Delta f^{\text{real}}(s_{\text{real}}^\xi, a_t') \approx \Delta f^{\text{sim}}(s_{\text{sim}}^\xi, a_t') + J_{a_t'}^{\text{sim}} G_\theta(\xi, (a_t', p)) \approx \Delta f^{\text{sim}}(s_{\text{sim}}^\xi, a_t^{\text{real}})$ . With the approximation of Equation (25) gives

$$J_{a_t'}^{\text{sim}} G_\theta(\xi, y_t') \approx \Delta f^{\text{sim}}(s_{\text{sim}}^\xi, a_t^{\text{real}}) - \Delta f^{\text{sim}}(s_{\text{sim}}^\xi, a_t'),$$
$$\approx J_{a_t'}^{\text{sim}}(a_t^{\text{real}} - a_t'), \tag{26}$$

yields $a_t' - a_t^{\text{real}} \approx -G_\theta(\xi, y_t')$. The $a_t'$ could be efficiently calculated with just a few steps of gradient descent, that gradients are only required to flow through Trunk Net only, leading to the real-world correction rule

$$a_t^{\text{real-corr}} = a_t^{\text{real}} - G_\theta(\xi, y_t'). \tag{27}$$

**Unified residual-action interpretation**  Equations equation 23 and equation 27 yield a consistent, domain-symmetric residual-action formulation:

$$\text{Simulation: } a_t^{\text{sim-corr}} = a_t + G_\theta(\xi, y_t), \qquad \text{Real: } a_t^{\text{real-corr}} = a_t^{\text{real}} - G_\theta(\xi, y_t')$$

Thus, the opposite signs arise naturally:

- **In simulation:** we *add* the residual to emulate missing real-world dynamics.

- **In reality:** we *subtract* the same residual to cancel hardware-specific biases and match the nominal simulator behavior.

### A.7.7  ROLE OF THE BRANCH-TRUNK DECOMPOSITION

We analyze how the Branch-Trunk decomposition adapts to different inputs through the following experimental setup: we randomly sample initial joint positions, velocities, and action sequences, execute the action sequences, and compare how the outputs of the Branch Net and Trunk Net vary with payload mass under the same initial state and action sequence. We also record how the Trunk Net output changes as each action in the sequence is executed under a fixed payload.

As shown in Figure 12, a clear trend emerges: compared to the baseline condition of 0kg payload, the deviations of both Branch Net and Trunk Net outputs from the baseline increase with payload mass. When the payload is held constant, the difference in Trunk Net outputs between consecutive timesteps remains statistically consistent throughout the action sequence. These Branch Net results demonstrate that our sensor model and Branch Net effectively **capture non-linear variations in system dynamics.**

Figure 13 compares the relative influence of payload on the Branch Net versus the Trunk Net. The results show that under the same change in payload, the deviation of the Branch Net output from the baseline is significantly larger than that of the Trunk Net—on average, the Branch Net variation is 7.9 times greater. Moreover, the effect of payload on the Trunk Net itself is relatively small compared to the effect of actions, accounting for only 20.35 % of the action-induced variation. This indicates that in **GapONet**, the **Branch Net primarily captures payload-dependent changes** in system dynamics, while the **Trunk Net focuses more on encoding action information, and remains payload-insensitive.**

Figure 14 illustrates the impact of payload mass on outputs for different joints. It can be observed that for both Branch Net and Trunk Net, the shoulder joints is most affected by payload, which aligns with the intuition that payload exerts a greater torque on the shoulder joint. Quantitatively, the influence of payload on the Trunk Net remains minimal compared to its effect on the Branch Net.

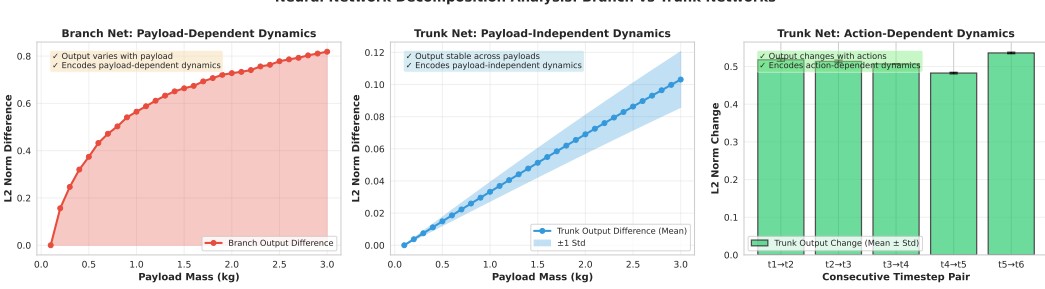

Figure 12: Variation of Branch Net and Trunk Net's values according to payload and action changes.

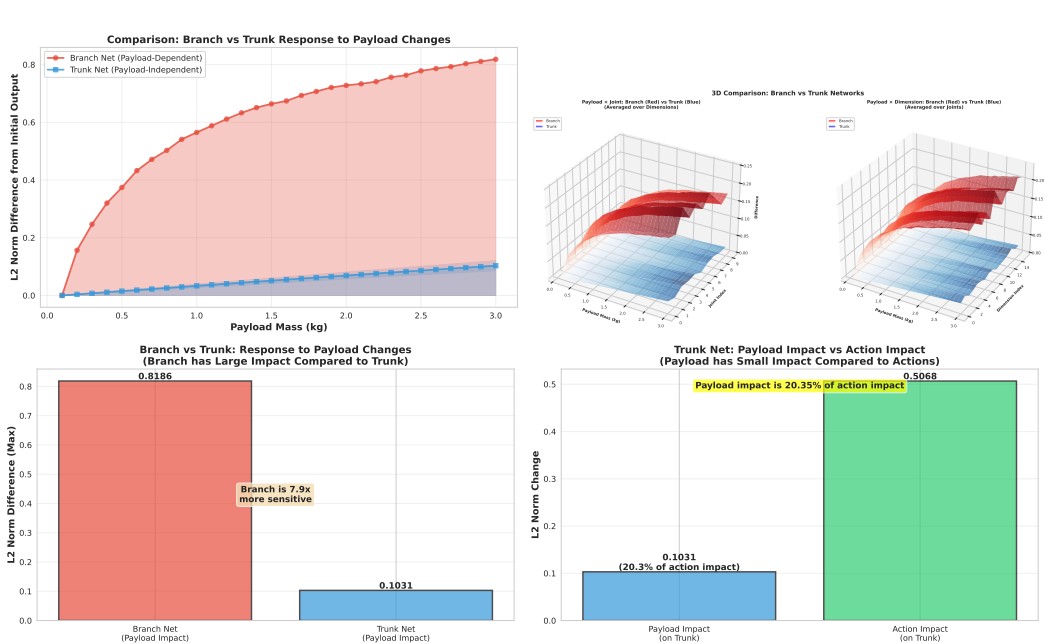

Figure 13: Comparison of payload and action's impact on Branch Net and Trunk Net.

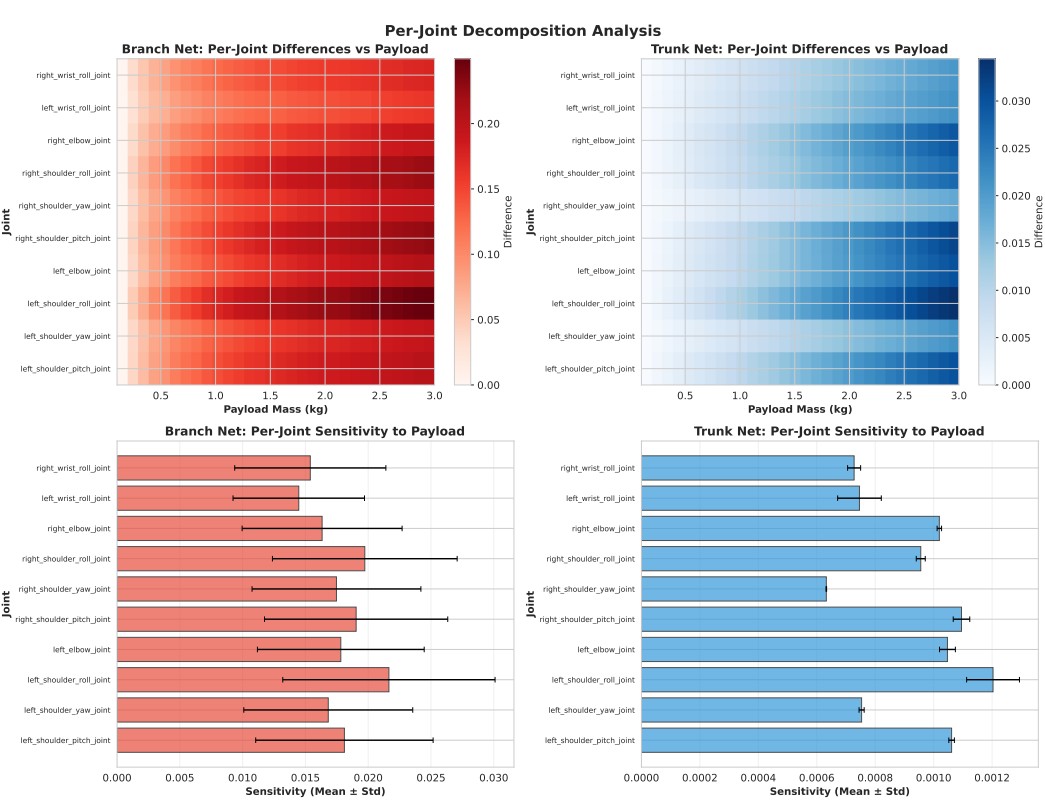

Figure 14: Impact of payload on Branch Net and Trunk Net's outputs on different joints.

## A.8 IMPLEMENTATION DETAILS

### A.8.1 NETWORK STRUCTURE

**Overview.** The training pipeline with `GapONet` consists of three components: a Sensor Predictor to predict the sensor input of Branch Network, a Branch Network $\mathcal{B}(U_q(x))$ that encodes sensor-driven actuation functions and a Trunk Network $\mathcal{T}(y)$ that processes action queries. Both are implemented as multi-layer perceptrons (MLPs), fused via dot product to yield the operator output $\mathcal{G}(U_q(x))(y)$. These networks are trained end-to-end with Proximal Policy Optimization (PPO), and optimized using Adam.

**Sensor Predictor**

- **Input:** For each time $j$ at time step $t$, the Sensor Predictor receives a sequence of sensor states over a h-step history window:

$$\{q_j^{t-n}, \dot{q}_j^{t-n}, q_{j,d}^{t-n}\}_{n=0}^{h},$$

  where $q_j, \dot{q}_j$ denote joint position and velocity, $q_{j,d}$ is the target position.
- **History Length:** $h = 4$
- **Input Dimension:** 10 joint num $\times$ (3 $\times$ history length + 1 current position) = 130-dim vector
- **Output:** $\Delta q \& \Delta \dot{q} \times 10$ joint = 20-dim vector
- **Sensor Number:** 20
- **Learning Rate:** $1 \times 10^{-4}$

**Branch Net.**

- **Input:** 20-dim vector of sensor predictor output $\times$ 20 sensor num = 400-dim vector
- **Delta Action Duration:** 1 step
- **Architecture:** 3-layer MLP with hidden sizes $[256, 256, 256]$, each followed by ELU activation.
- **Output:** $p$-dimensional latent representation ($p = 160$, *i.e.* $16 \times 10 = \text{num\_basis} \times \text{num\_actions}$ by default)
- **Learning Rate:** $1 \times 10^{-4}$

**Trunk Net.**

- **Input:** The Trunk Net receives the target query $y = q_{j,d}^{t+1}$ desired joint position + payload
- **Input Dimension:** 11
- **Architecture:** 3-layer MLP with hidden sizes $[128, 128, 128]$, ELU activations
- **Output:** $p$-dimensional vector, same dimension as Branch output
- **Learning Rate:** $1 \times 10^{-4}$

**Fusion.** The operator output is computed as the dot product:

$$\mathcal{G}(U_q(x))(y) = \sum_{i=1}^{J} \mathcal{B}_i(x) \cdot \mathcal{T}_i(y),$$

where $J$ is the number of actuated joints. Specifically, we reshape both output of Branch Net and Trunk Net to $16 \times 10$, perform Hadamard product and then sum over the first dimension.

**Training Details.**

- PPO update with clipping ratio $\epsilon = 0.2$, batch size = 4096.

- Reward defined as $r_t = -\|q^{t+1} - q_{\text{real}}^{t+1}\|^2$.

- Temporal smoothness penalty $\mathcal{L}_{\text{gap}}$ with $\lambda = 0.01$.

- Training duration: 1 hour on 1 RTX 3090Ti GPU.

Table 6: Hyperparameters for Branch Net.

| Hyper-Parameters | Values |
|---|---|
| History Length | 4 |
| Delta Action Duration | 1 |
| Sensor Number | 20 |
| $U_q$ Input | $A, V, P, J$ |
| $U_q$ Output | $\Delta S$ |
| Layer Structure | $[256, 256, 128]$ |
| Output Number | 10 |
| Dropout | 0.1 |
| Samples Per Update Iteration | 131072 |
| Policy/Value Function Minibatch Size | 16384 |
| Discriminators/Encoder Minibatch Size | 4096 |
| $\gamma$ Discount | 0.99 |
| Learning Rate | $2 \times 10^{-5}$ |
| GAE($\lambda$) | 0.95 |
| TD($\lambda$) | 0.95 |
| PPO Clip Threshold | 0.2 |
| $T$ Episode Length | 300 |

### A.8.2 SIMULATIONS

We evaluate on MuJoCo 3.2.3, Isaac Gym 1.0rc4, and Isaac Sim 4.5.0. To enhance reproducibility, each setting uses the simulator's official default parameters. The software environments are:

- MuJoCo / Isaac Gym: Python 3.8.13, legged_gym 1.0.0, PyTorch 2.4.1, torchvision 0.19.1.

- Isaac Sim: Python 3.10.4, isaaclab 0.40.21, PyTorch 2.5.1, torchvision 0.20.1.

### A.8.3 BASELINES

**PD control**    As shown in Section A.4.1, we employ PD control to drive the humanoid robot in both simulation and the real world. In the simulator, we use the `ImplicitActuator` API in IsaacLab to compute the applied torque from the input action. For real hardware, we rely on the official APIs provided by the Unitree and RealMan humanoid platforms to obtain the torque computed by their onboard PD controllers. The corresponding implementation details and code *real_robot_deploy.py* are included in the supplementary materials for reference.

Table 7: Hyperparameters for Trunk Net.

| Hyper-Parameters | Values |
|---|---|
| History Length | 4 |
| Delta Action Duration | 1 |
| Sensor Number | 20 |
| $y$ Input | $a_d$ |
| Layer Structure | $[128, 128]$ |
| Output Number | 10 |
| Dropout | 0.1 |
| Samples Per Update Iteration | 131072 |
| Policy/Value Function Minibatch Size | 16384 |
| Discriminators/Encoder Minibatch Size | 4096 |
| $\gamma$ Discount | 0.99 |
| Learning Rate | $2 \times 10^{-5}$ |
| GAE($\lambda$) | 0.95 |
| TD($\lambda$) | 0.95 |
| PPO Clip Threshold | 0.2 |
| $T$ Episode Length | 300 |

**MLP**   For the MLP baseline, we follow the approach used in He et al. (2025). Specifically, the collected sim–real paired data are fitted with an MLP to learn a mapping from the simulated action to the real-world delta action. The model adopts a standard Actor–Critic architecture, where both the actor and critic networks use a [1000,200] MLP with ELU activations. Training is conducted using PPO Schulman et al. (2017), and the hyperparameters are summarized in Section A.8.3.

Table 8: Hyperparameters for PPO training in MLP baseline.

| Hyper-Parameters | Values |
|---|---|
| Value loss coef | 1.0 |
| Clip parameter | 0.2 |
| Entropy coef | 0.0 |
| Learning epochs | 5 |
| Mini batches | 4 |
| Learning rate | $1 \times 10^{-4}$ |
| Schedule | adaptive |
| $\gamma$ Discount | 0.99 |
| Desired KL | 0.008 |
| Environments | 4096 |
| Number of steps in each env | 32 |

**Transformer**   The Transformer baseline follows the same PPO training setup as the MLP baseline He et al. (2025), with the only difference being the replacement of the actor–critic MLP with a Transformer-based architecture. The hyperparameters used for training are identical to those of the MLP baseline, as shown in Section A.8.3. We also implement a Transformer-based baseline using an Actor–Critic architecture. The observation (250-dimensional) is first projected to a 128-

dimensional embedding, followed by a two-layer Transformer encoder with $d_{model} = 128$, four attention heads, feedforward dimension 512, and GELU activation. The actor maps the encoded feature to a 10-dimensional Gaussian action distribution (with a learnable scalar log-std), while the critic shares the same encoder and outputs a scalar value.

**Domain Randomization** We adopt the motion-tracking policy widely used in industry Liao et al. (2025). Since the original policy was trained on the Unitree G1 robot, we replace the URDF and related configuration files with those of the H1-2 platform and retrain the motion tracker using imitation learning. The hyper-parameter of humanoid body is calculated by System Identification in the next prargraph. To better align with our paper's setting involving varying payloads, we additionally apply domain randomization on the payload: during imitation learning, the payload mass is randomized by sampling from 0, 1, 2, 3. This improves the robustness and stability of the tracker under different payload conditions. The reward terms used for training our tracker are listed in Section A.8.3:

Table 9: Reward formulation for training tracker with domain randomization.

| Reward Terms | Equation | Weights |
|---|---|---|
| Body Position | $\exp\left(-\left(\frac{1}{\|\mathcal{B}_{\text{target}}\|}\sum_{b\in\mathcal{B}_{\text{target}}}\frac{\|\mathbf{p}_b^{\text{des}}-\mathbf{p}_b\|^2}{0.3^2}\right)\right)$ | 1.0 |
| Body Orientation | $\exp\left(-\left(\frac{1}{\|\mathcal{B}_{\text{target}}\|}\sum_{b\in\mathcal{B}_{\text{target}}}\frac{\|\log(R_b^{\text{des}}R_b^\top)\|^2}{0.4^2}\right)\right)$ | 1.0 |
| Body Linear velocity | $\exp\left(-\left(\frac{1}{\|\mathcal{B}_{\text{target}}\|}\sum_{b\in\mathcal{B}_{\text{target}}}\frac{\|\mathbf{v}_b^{\text{des}}-\mathbf{v}_b\|^2}{1.0^2}\right)\right)$ | 1.0 |
| Body Angular velocity | $\exp\left(-\left(\frac{1}{\|\mathcal{B}_{\text{target}}\|}\sum_{b\in\mathcal{B}_{\text{target}}}\frac{\|\boldsymbol{\omega}_b^{\text{des}}-\boldsymbol{\omega}_b\|^2}{3.14^2}\right)\right)$ | 1.0 |
| Anchor Position (Optional) | $\exp\left(-\frac{\|\mathbf{p}_{\text{anchor}}^{\text{des}}-\mathbf{p}_{\text{anchor}}\|^2}{0.3^2}\right)$ | 0.5 |
| Anchor Orientation (Optional) | $\exp\left(-\frac{\|\log(R_{\text{anchor}}^{\text{des}}R_{\text{anchor}}^\top)\|^2}{0.4^2}\right)$ | 0.5 |

**System Identification** We follow the standard practice of locally linearizing the joint-space dynamics around collected motion trajectories. Under the manipulator equation Ortega et al. (1998): Ortega et al. (1998):

$$\tau = M(q)\ddot{q} + C(q,\dot{q})\dot{q} + g(q). \tag{28}$$

Given the position error $e_{\text{pos}} = q^{\text{cmd}} - q$ and velocity error $e_{\text{vel}} = \dot{q}$, we fit an affine model $\tau \approx k_p e_{\text{pos}} + k_d e_{\text{vel}} + b$, using ordinary least squares (scikit-learn `LinearRegression`). The input feature matrix is $X = [e_{\text{pos}}, e_{\text{vel}}] \in \mathbb{R}^{N \times 2}$ and the target is the measured joint torque $Y \in \mathbb{R}^N$. We estimate $(k_p, k_d, b)$ using ordinary least squares. This yields a classical linear system-identification baseline that captures the best local linear approximation to the underlying dynamics. Training is instantaneous, as the solution is obtained via analytical least-squares minimization.

**Network-based System Identification** We further approximate the joint dynamics using a multi-layer perceptron (MLP) Hwangbo et al. (2019), which learns a flexible nonlinear function

$$\tau = f_{\text{MLP}}(e_{\text{pos}}, e_{\text{vel}}). \tag{29}$$

The MLP consists of two hidden layers of sizes $(100, 50)$ with ReLU activations and is trained using the Adam optimizer for up to 1000 iterations (`MLPRegressor`, `max_iter=1000`, `activation=relu`, `solver=adam`, `random_state=42`). The trained model captures frictional, configuration-dependent, and actuator nonlinearities. For interpretability and fair comparison to linear baselines, we optionally project the MLP predictions onto a PD-like form via: $f_{\text{MLP}}(e_{\text{pos}}, e_{\text{vel}}) \approx k_p e_{\text{pos}} + k_d e_{\text{vel}} + b$. We additionally monitor the optimization status (final loss value and number of iterations used) to ensure convergence and report the resulting $R^2$ score on the training dataset.

**Kernel-based System Identification**  To capture nonlinear components of the joint dynamics—such as friction, motor response nonlinearities, and configuration-dependent coupling—we employ Support Vector Regression (SVR) with an RBF kernel Deisenroth et al. (2013) to model the mapping

$$\tau = f_{\text{SVR}}(e_{\text{pos}}, e_{\text{vel}}). \tag{30}$$

We train the regressor using scikit-learn's `SVR` implementation with default hyperparameters (`kernel=rbf`, $C$=1.0, $\epsilon$=0.1, $\gamma$=scale). As with the MLP baseline, we optionally obtain PD-like gains via linear projection of the predicted torques: $f_{\text{SVR}}(e_{\text{pos}}, e_{\text{vel}}) \approx k_p e_{\text{pos}} + k_d e_{\text{vel}} + b$. Kernel methods provide strong nonlinear regression behavior while maintaining good sample efficiency. This two-stage process yields interpretable rigid-body parameters $(k_p, k_d, b)$ while allowing the SVR to model nonlinear torque dependencies. All training uses the same feature matrix $X$ and target vector $Y$ as in the linear baseline. The coefficient of determination $R^2$ is computed to quantify the quality of the nonlinear fit prior to linear projection.

**Lower-body Locomotion Policy.**  To conduct the Locomotion Trajectory Tracking experiment, we trained a lower-body locomotion policy capable of stable walking. The policy is adapted from HOMIE Ben et al. (2025) with modifications for the H1-2 platform, including updates to the URDF and the reward design. The full reward formulation and corresponding weights are listed in Section A.8.3:

Table 10: Reward formulation for lower-body locomotion policy.

| Reward Terms | Equation | Weights |
|---|---|---|
| x Vel. tracking | $\exp\left\{-4\left\|v_x - v_{r,x}\right\|_2^2\right\}$ | 1.5 |
| y Vel. tracking | $\exp\left\{-4\left\|v_y - v_{r,y}\right\|_2^2\right\}$ | 1.0 |
| Ang. Vel. tracking | $\exp\left\{-4\left\|\omega_{\text{yaw}} - \omega_{r,\text{yaw}}\right\|_2^2\right\}$ | 1.0 |
| Base height tracking | $\exp\left\{-4\left\|h_t - h_{r,t}\right\|_2^2\right\}$ | 2.0 |
| Lin. Vel. z | $v_{r,z}^2$ | -0.5 |
| Ang. Vel. xy | $\left\|\omega_{r,xy}\right\|_2^2$ | -0.025 |
| Orientation | $\left\|g_x\right\|_2^2 + \left\|g_y\right\|_2^2$ | -1.5 |
| Action rate | $\left\|a_t - a_{t-1}\right\|_2^2$ | -0.01 |
| Hip joint deviation | $\sum_{\text{hip joints}} \left\|\theta_i - \theta_i^{\text{default}}\right\|^2$ | -0.5 |
| Ankle joint deviation | $\sum_{\text{ankle joints}} \left\|\theta_i - \theta_i^{\text{default}}\right\|^2$ | -0.75 |
| Squat knee | $-\left\|\left(h_{r,t} - h_t\right)\left(\frac{q_{\text{knee},t} - q_{\text{knee,min}}}{q_{\text{knee,max}} - q_{\text{knee,min}}} - \frac{1}{2}\right)\right\|$ | -0.75 |
| DoF Acc. | $\sum_{\text{all joints}} \left\|\frac{\dot{q}_{t,i} - \dot{q}_{t-1,i}}{dt}\right\|_2^2$ | $-2.5 \times 10^{-7}$ |
| DoF pos limits | $\sum_{\text{all joints}} \text{out}_i$ | -2.0 |
| Feet air time | $\mathbb{1}_{\{\text{first contact}\}}(T_{\text{air}} - 0.5)$ | 0.05 |
| Feet clearance | $\sum \left(p_z^{\text{target}} - p_z^i\right)^2 \cdot v_{xy}^i$ | -0.25 |

## A.9  LIMITATION AND FUTURE WORK

Our dataset and analysis primarily target the upper body, and although we include tests on locomotion trajectory tracking, the present system does not yet enable highly dynamic sim–real transfer for full humanoids. Going forward, we will (i) extend the current pipeline to high-dynamics, whole-body loco-manipulation and to additional robot platforms, and (ii) address the strong dependence on a stable locomotion policy—even with relative metrics, unreliable gaits can cause catastrophic failures (cf. 'videos/failure.mp4') that preclude testing. A second focus is to train a robust full-body tracker for large-mass humanoids (e.g., H1-2), providing a stronger substrate for our operator-based sim–real mapping.

