# OpenReview forum: "GapONet: Nonlinear Operator Learning for Bridging the Humanoid Sim-to-Real Gap"
_ICLR.cc/2026/Conference — Submitted to ICLR 2026_

### Official Review · Reviewer_DFMY · 2025-10-31

**Soundness:** 3
**Presentation:** 2
**Contribution:** 2
**Rating:** 2
**Confidence:** 3

**Summary:**

The paper presents a well-motivated study on improving sim-to-real transfer for humanoid robots. It contributes a sim–real data collection pipeline and introduces TWINS, the first dataset focusing on payload-induced domain gaps across multiple robots, motions, and simulators. The authors further provide 30+ hours of synchronized sim–real data and quantitative analyses of simulator discrepancies. Finally, they propose GapONet, a payload-conditioned nonlinear operator that maps simulation actuation functions to residual hardware actions, demonstrating its feasibility through reinforcement learning.

**Strengths:**

1. The paper addresses a highly practical challenge in humanoid sim-to-real transfer by explicitly modeling the payload effect not as random noise but as a structured conditioning variable during motion tracking. This perspective is realistic and directly relevant for real-world humanoid deployment.
2. Building on DeepONet’s operator-learning framework, the paper presents a novel combination between nonlinear operator learning and the sim-to-real transfer process in robotics. It highlights the separable nature of the payload by treating it as the query variable in the operator formulation. Moreover, it introduces an insightful perspective on sim-to-real alignment—modeling the domain gap as a mapping from simulation actuation functions to residual actions, rather than as pointwise corrections.
3. The paper provides detailed technical descriptions of its data collection pipeline, including the construction of a cross-robot, cross-simulator dataset covering diverse humanoid motions under varying payload conditions. The dataset curation and labeling process are thoroughly documented.
4. The paper also includes a joint-level comparison across three mainstream simulators. Although not comprehensive and is not the focus of this paper, this analysis provides useful insights and contributes valuable reference for the community.

**Weaknesses:**

1. The paper does not compare the proposed method with other nonlinear system identification approaches, such as neural network-based or kernel-based methods. The related work section on nonlinear system identification is insufficient.
2. The generalization capability of GapONet beyond payload variation remains unclear. As a method for bridging the sim-to-real gap, its applicability under other changing factors is not demonstrated.
3. The experimental settings lack clarity, e.g. the training details of baseline methods are missing, including the choice of training and test sets, which makes reproducibility difficult.
4. The training pipeline and design choices are not well explained. It is unclear why reinforcement learning was chosen.
5. Some mathematical symbols appear in the paper without explicit definitions, making the derivations harder to follow.

**Questions:**

1. In Figure 1, why does the dataset illustration show the Unitree G1 performing Kung Fu, while the data used for experiments only include three types of lower-body gaits without whole-body tracking motions?
2. Why did you choose a reinforcement learning (PPO) algorithm to train GapONet, given that the original DeepONet paper uses supervised learning? The paper does not explain the motivation or provide the formal RL formulation.
3. In Section 4.3, you claim that computing all sensor values is computationally prohibitive. Could you clarify whether this limitation is due to the RL setup? If so, why is RL used instead of supervised regression?
4. Why is the default large-gap ratio set to 0.5 rad, especially when Figure 3(b) shows that the typical error is below 0.3 rad?
5. In Section 3.3.2, you state that the deviation is nonlinear and that phase lag is related to payload. However, the phase difference between sim and real does not necessarily imply delay, and there is no quantitative result showing nonlinearity. Could you clarify this claim?
6. How did you select the data and motions used in Section 3.3? In Figure 3(b), the deviation at 0 kg payload reaches 0.1 rad, but in Figure 3(c), the inter-simulator joint angles appear much smaller. Why?
7. In the motion tracking and trajectory tracking experiments, what role does GapONet play in real-world tests?
8. In Section 5.2, you mention an online residual compensation method. Could you describe this method in detail? It is currently unexplained.
9. In Section 5.1(ii), what is the "Transformer-learned dynamics model"? Please specify its structure and training configuration.

Additional Feedback:

1. Writing and formatting:
   - Redundant sentence in Section 2.2.
   - Uneven layout and spacing issues in Figure 1.
   - Multiple typos in the *Method* section.
2. Technical presentation:
   - Please define all mathematical symbols when they first appear.
   - Consider reorganizing Section 4 to make the training pipeline easier to follow.
   - Provide explicit details for experimental setup.

---

> ### Author Response · Authors · 2025-11-21
> **Response to the Reviewer (1 / 3)**
>
> We sincerely appreciate your positive assessment of our work as "novel" and your acknowledgment that it "provides useful insights and contributes valuable reference." We are also grateful for your careful suggestions on writing, formatting, and supplementary experiments, which have helped us enhance the clarity and quality. Regarding the weaknesses, questions, and feedback you raised, we have prepared a point-by-point response below:
>
> ### **[W1 Compare with Nonlinear System Identification]**
> As shown in Appendix A.8.3, we implement system-identification baselines by fitting rigid-body dynamics models [1,2]. Follow your suggestion to further evaluate whether system identification can address the payload-induced nonlinearities, we additionally include MLP-based [3] and kernel-based [4] system ID methods, with results reported in **Table 1 and Appendix A.7.4(table below)**. Across all methods, the identified parameters change only slightly, and linear and nonlinear models perform almost identically, indicating that system ID mainly tunes static hyperparameters and is insufficient for capturing payload-dependent nonlinear effects. This supports our conclusion that real-time correction of action and torque is necessary beyond traditional system identification. Detailed implementation and analysis are provided in **Appendix A.7.4 and A.8.3**.
>
> | Method                | LGR (%) ↓ (0 kg) | IQR ↓ (0 kg)      | Range ↓ (0 kg)    | LGR (%) ↓ (1 kg) | IQR ↓ (1 kg)      | Range ↓ (1 kg)    |
> | --------------------- | ---------------- | ----------------- | ----------------- | ---------------- | ----------------- | ----------------- |
> | System Identification | 12.4 ± 0.3       | 0.141 ± 0.015     | 0.505 ± 0.032     | 9.01 ± 1.0       | 0.140 ± 0.029     | 0.609 ± 0.122     |
> | Network-based SysID   | 12.5 ± 0.06      | 0.154 ± 0.019     | 0.441 ± 0.001     | 13.1 ± 0.65      | 0.129 ± 0.031     | 0.538 ± 0.002     |
> | Kernel-based SysID    | 13.3 ± 0.14      | 0.155 ± 0.019     | 0.497 ± 0.006     | 8.84 ± 2.37      | 0.129 ± 0.015     | 0.588 ± 0.002     |
> | **Ours (GapONet)**    | **0.09 ± 0.03**  | **0.093 ± 0.016** | **0.449 ± 0.117** | **0.22 ± 0.11**  | **0.115 ± 0.013** | **0.537 ± 0.148**     |
>
> [1] Ortega R, Loria A, Nicklasson P J, et al. Euler-Lagrange systems[M]//Passivity-based Control of Euler-Lagrange Systems: Mechanical, Electrical and Electromechanical Applications. London: Springer London, 1998: 15-37.
> [2] Sousa C D, Cortesao R. Physical feasibility of robot base inertial parameter identification: A linear matrix inequality approach[J]. The International Journal of Robotics Research, 2014, 33(6): 931-944.
> [3] Hwangbo J, Lee J, Dosovitskiy A, et al. Learning agile and dynamic motor skills for legged robots[J]. Science Robotics, 2019, 4(26): eaau5872.
> [4] Deisenroth M P, Fox D, Rasmussen C E. Gaussian processes for data-efficient learning in robotics and control[J]. IEEE transactions on pattern analysis and machine intelligence, 2013, 37(2): 408-423.
>
> ### **[W2 The Generalization Capability of GapONet]**
> In addition to payload changes, our evaluation already includes two other forms of variation: unseen motions and unseen hardware. All test trajectories are zero-shot motions not used in training（clarified in **Lines 403–405**）and we added t-SNE visualizations of the train/test distributions (position, velocity, and torque) in **Supplementary Appendix A.3.3**. The visualization shows that the distributions barely overlap, further demonstrating that GapONet can generalize across both payloads and motions.
>
> Moreover, as stated in Line 474 and in the captions of Figure 1 and Table 2, all evaluations are performed on a brand-new robot that was never used during data collection. This confirms that our method generalizes across hardware instances with small manufacturing differences.

---

> > ### Author Response · Authors · 2025-11-21
> > **Response to the Reviewer (2 / 3)**
> >
> > ### **[W3 & W4 & W5 & Q9 Mathematical symbols and details]**
> > Thank you for the detailed suggestions. We have addressed the issues you raised. All revisions are highlighted in blue in the updated draft.
> > Specifically:
> > - We revised Section 2.2 to improve clarity, and added a brief discussion of nonlinear system identification in Section 2.1.
> > - We fixed the formatting issue in Figure 1 and updated the draft.
> > - We reorganized Section 4 for better readability, added necessary details to clarify the training pipeline, corrected typos, and ensured that all symbols are clearly defined upon first appearance.
> > - We added explanations in Appendix A.3.3 (train/test data split), Section 5 & Appendix A.7.6 (Experiment setup), and Appendix A.8.3 (baseline execution details).
> > - To further improve readability within space limits, we added pseudocode for the method in Appendix A.6 and provided a complete list of symbol definitions in Appendix A.6.2.
> >
> > ### **[W4 & Q2 & Q3 Why Chose Reinforcement Learning]**
> > Our objective depends on non-smooth simulator dynamics (contact, actuator nonlinearities, friction). While differentiable simulators exist, they generally lack GPU-parallel execution and are orders of magnitude slower for humanoid-scale training[1,2]. Thus, supervised learning would require backpropagating through Isaac Gym/Isaac Sim, which is infeasible.
> > RL avoids this issue by treating the simulator as a black-box transition model and optimizing the operator directly from trajectory-level rewards. This makes RL the only practical and efficient choice for training under non-differentiable sim-to-real dynamics. We have provided the formal RL formulation in Section 4.3. Further details appear in **Appendix A.6.3**.
> >
> > [1] de Avila Belbute-Peres F, Smith K, Allen K, et al. End-to-end differentiable physics for learning and control[J]. Advances in neural information processing systems, 2018, 31.
> > [2] Hu Y, Anderson L, Li T M, et al. Difftaichi: Differentiable programming for physical simulation[J]. arXiv preprint arXiv:1910.00935, 2019.
> >
> > ### **[Q1 Where is G1's Whole-body Tracking]**
> > We implemented whole-body tracking on the G1 and included **additional motion videos in the supplementary materials.** However, due to G1’s small form factor and lightweight design, operating with two dexterous hands under payload introduces substantial motor load. In indoor conditions, the motors overheat after ~1 minute, making it impractical to collect sufficient high-quality data for the payload experiments. For this reason, we chose not to include these limited samples in the main analysis.
> > The complete whole-body pipeline will be released in our upcoming codebase to support multi-platform use and future studies as stronger hardware becomes available.
> >
> > ### **[Q4 Why Chose 0.5 rad as the Threshold of LGR]**
> > As for the metric, there is currently no standardized measure for humanoid sim-real evaluation. We therefore follow the definition in [1] and use the proportion of frames whose joint errors fall outside a tolerance—our LGR. Motions are aligned by timestamp, and we compute joint-wise errors between sim and real. Since the sim-real gap can never be eliminated (even repeated replays differ), LGR provides a practical measure of how well the discrepancy is kept small.
> > We adopt the commonly used 0.5 rad threshold, which prior work [2,3] employs as a perturbation magnitude for identifying severe tracking failures rather than normal fluctuations. This value is intentionally set far above typical joint-tracking errors in robot control. Rrrors exceeding 0.5 rad correspond to catastrophic sim-to-real failures, making LGR a meaningful indicator of such gaps. The detailed reason why we chose 0.5rad can be referred to **Appendix A.7.1**.
> >
> > [1] Li S, et al. Hand pose estimation for hand-object interaction cases using augmented autoencoder[C]. ICRA. IEEE, 2020: 993-999.
> > [2] Zhang J, et al. Fuzzy fractional-order PID control for two-wheeled self-balancing robots on inclined road surface[J]. Systems Science & Control Engineering, 2022, 10(1): 289-299.
> > [3] Sun Y. Automatic vibration control method for grasping end of flexible joint robot[J]. Journal of Vibroengineering, 2023, 25(8): 1502-1515.

---

> ### Author Response · Authors · 2025-11-21
> **Response to the Reviewer (3 / 3)**
>
> ### **[Q4 Why Figure 3(b) Shows that the Typical Error is Below 0.3 rad]**
> As Q4 response [Why choose 0.5 rad as the LGR threshold], 0.5 rad is a value far above typical joint-tracking errors in robot control. Such large gaps rarely appear in randomly collected data—otherwise the dataset itself would be unreliable. The distributional analysis of the sim–real gap in the TWINS dataset, together with the corresponding visualizations, in **Appendix A.7.1 and Figure 10.**
>
>
> ### **[Q5 The Phase and Nonlinearity between Sim and Real]**
> We agree that a phase difference between sim and real does not necessarily imply delay. In our paper, the statement in line 281 was only intended to report the empirical observation shown in Fig. 3(b)—that different payloads introduce visible phase shifts in practice. The actual evidence for nonlinear growth of the sim–real gap comes from Fig. 2(b), where different joints exhibit non-uniform and non-monotonic increases as payload increases. We have revised the text around line 281 to clarify this point.
>
> ### **[Q6 Different Gap in Figure3(b) & Figure3(c)]**
> Fig. 3(c) reports sim–sim discrepancies: although the simulators use similar rigid-body dynamics, variations in contact models and numerical schemes still produce differences. However, these remain smaller than the sim–real gap in Fig. 3(b), where actuator friction, compliance, latency, and other unmodeled nonlinearities dominate; the difference in magnitude is expected.  The much larger sim–real discrepancy directly highlights the importance of explicitly learning this gap. All data and motions are randomly sampled to reflect the dataset’s overall characteristics rather than cherry-picking atypical examples.
>
>
> ### **[Q7 & Q8 The Role of GapONet in real-world test]**
> The “online residual compensation” refers to the role that GapONet plays during real-robot deployment (Experiment 'locomotion trajectory tracking'). We have added **a brief clarification of this procedure in Section 5.2 and detail analysis in Supplementary Appendix A.7.6.**
>
> We remain open to further discussion and are happy to supply any additional information or technical details as needed. Should our revisions satisfactorily resolve the issues highlighted, we would be deeply appreciative if you could consider a higher score. Your insightful feedback has been invaluable in improving this work, and we thank you once again for your time and constructive input.

---

### Official Review · Reviewer_unGq · 2025-11-01

**Soundness:** 3
**Presentation:** 2
**Contribution:** 3
**Rating:** 6
**Confidence:** 3

**Summary:**

This paper studies sim-to-real gaps introduced due to the end-effector payload during object interaction for humanoid loco-manipulation tasks. The authors proposed TWINS, the first dataset focused on payload-induced sim-to-real gaps across multiple robots, and found a consistent nonlinear increase in dynamics error. They address this by using GapONet, a nonlinear operator that maps simulation context features to real with a function-to-function learning objective, to propose delta actions that compensate for such gaps. Through empirical studies, they demonstrate the effectiveness of their method in motion tracking tasks.

**Strengths:**

This paper is well motivated, it provides solid problem formulation and theoretically analysis.

**Weaknesses:**

There exist multiple typos in this manuscript (for example: “Modelin” in figure 1) and inconsistent font. These errors in the main figure may reduce reader confidence in the presentation.

**Questions:**

1. "The discrepancy arises from coupled channels—gravity, friction, Coriolis and inertial coupling, actuator limits and efficiency drift, sensing noise, and delays—that a pointwise function mapping cannot capture or generalize." I am still not convinced why learning operators of actuator functions has been necessary or superior to point-wise mappings. Can authors provide more empirical evidence that the insufficiency of the point-wise mapping method fails to generalize?

2. In Table 1, while all methods demonstrate relatively close IQR and Range, baseline methods have significantly larger LGR. What is the explanation of this difference? I am also curious about the effectiveness of these metrics. Which one of the does the author consider the most faithful in measuring the sim-to-real gap?

3. How does the proposed method work for motion tracking that includes agility or actual object interaction? Motion tracking with payload might not be

---

> ### Author Response · Authors · 2025-11-21
> **Response to the Reviewer**
>
> We sincerely thank you for acknowledging the strength of our formulation and theoretical analysis, and we appreciate your careful identification of typos. We also value your requests for more detailed experimental explanations and suggestions for further improvements. Our detailed responses are as follows:
>
> ### **[W1 Typo in Figure 1]**
> We have fixed the problem of line breaks and updated the draft accordingly.
>
> ### **[Q1 Why Learning Operators of Actuator Functions]**
> Your point is valuable—providing more empirical evidence that point-wise mappings fail to generalize indeed strengthens the justification for our approach. In response, we added the comparison in **Appendix A.7.3**, evaluating a standard high-capacity MLP conditioned on both payload and point-wise simulation context. The results show that under zero-shot motions, the MLP exhibits significantly poorer generalization than the operator-based method, producing large gap ranges.
>
> In addition, we further examined the effects of model capacity and history information, resulting in **nine new experiments along with their corresponding error curves**. Please refer to **Supplementary Appendix A.7.3** for the detailed analysis.
>
>
> ### **[Q2 Baseline method's LGR Explanation]**
> **Close IQR & Range, while larger LGR:** Although the IQR and Range in Table 1 appear similar across methods, this is expected: both metrics measure overall error dispersion, whereas LGR is a threshold-based metric that is highly sensitive to the number of frames exceeding 0.5 rad (see Appendix A.7.1). Thus, baseline methods can show comparable IQR/Range yet still accumulate many large-error events, resulting in much higher LGR. **Meanwhile, IQR and Range remain useful.** IQR reflects typical-case stability, and Range quantifies rare but extreme deviations. Experiments in Appendix A.7.3 show that the Range grows to nearly 1.0 rad under other baselines, confirming that it is an effective measure when large discrepancies occur.
>
> As for the faithful metric, there is currently no standardized measure for humanoid sim-real evaluation. We therefore follow the definition in [1] and use the proportion of frames whose joint errors fall outside a tolerance—our LGR. Motions are aligned by timestamp, and we compute joint-wise errors between sim and real. Since the sim-real gap can never be eliminated (even repeated replays differ), LGR provides a practical measure of how well the discrepancy is kept small. We adopt the commonly used 0.5 rad threshold, which prior work [2,3] employs as a perturbation magnitude for identifying severe tracking failures rather than normal fluctuations. This value is intentionally set far above typical joint-tracking errors in robot control. Rrrors exceeding 0.5 rad correspond to catastrophic sim-to-real failures, making LGR a meaningful indicator of such gaps. The detailed reason why we chose 0.5rad can be referred to **Appendix A.7.1**.
>
> ### **[Q3 Motion Tracking with Object Interaction]**
> This is an important direction for our future work, and we appreciate the reviewer raising it. As an initial step toward true interactive manipulation, we include in the **supplementary materials demonstrations of the H1-2 performing motion-tracking-based lifting of a lightweight box and a heavier chair**. We are actively extending these results, and additional interactive scenarios will be explored in our follow-up work.
>
>
> We hope our responses and additional experimental results have fully addressed your concerns. We remain open to further discussion and are happy to provide any additional information as needed. If our revisions have satisfactorily resolved the issues raised, we would be very grateful for your consideration of a higher score. Your insightful feedback has been essential in improving our work, and we thank you once again for your time and constructive comments.
>
>
> [1] Li S, et al. Hand pose estimation for hand-object interaction cases using augmented autoencoder[C]. ICRA. IEEE, 2020: 993-999.
>
> [2] Zhang J, et al. Fuzzy fractional-order PID control for two-wheeled self-balancing robots on inclined road surface[J]. Systems Science & Control Engineering, 2022, 10(1): 289-299.
>
> [3] Sun Y. Automatic vibration control method for grasping end of flexible joint robot[J]. Journal of Vibroengineering, 2023, 25(8): 1502-1515.

---

### Official Review · Reviewer_R4BQ · 2025-11-01

**Soundness:** 2
**Presentation:** 3
**Contribution:** 2
**Rating:** 4
**Confidence:** 3

**Summary:**

This paper, "GapONet: Nonlinear Operator Learning for Bridging the Humanoid Sim-to-Real Gap," addresses the critical challenge of transferring policies learned in simulation to real-world humanoid robots. The authors correctly identify that the sim-to-real gap is exacerbated by complex, payload-induced nonlinearities and unmodeled dynamics in high-DoF systems. To tackle this, they propose GapONet, a novel payload-conditioned nonlinear operator network. GapONet is designed to learn a mapping from the simulation context function (i.e., the state and action in sim) to the residual action required on the real hardware, effectively acting as a nonlinear correction layer. The authors also introduce TWINS, a large-scale, synchronized sim-to-real dataset collected across multiple simulators and a real humanoid platform, which is a significant contribution in itself. Experimental results demonstrate that GapONet achieves superior performance in reducing the sim-to-real gap compared to competitive baselines, showing a reduction in tracking error and improved stability, particularly under varying payload conditions.

**Strengths:**

The core idea of framing the sim-to-real gap correction as a nonlinear operator learning problem is highly original and compelling. While prior work has used residual learning, the application of a payload-conditioned operator network (inspired by Neural Operators) to model the complex, functional relationship of the sim-to-real discrepancy is a novel approach in the context of humanoid robotics. The use of a branch-trunk decomposition to separate the payload-independent dynamics from the payload-dependent non-linearities is a clever architectural choice that enhances generalization. The work is highly significant for the field of sim-to-real transfer, especially for complex, high-DoF systems like humanoids. The introduction of the TWINS dataset is a valuable resource for future research. The GapONet model offers a powerful, generalizable framework for modeling complex unmodeled dynamics, which could be broadly applicable beyond payload variation to other sources of discrepancy (e.g., friction, compliance).

**Weaknesses:**

1. While the operator network formulation is the central claim of the paper, the experiments do not sufficiently justify its necessity over a standard, high-capacity Multi-Layer Perceptron (MLP) with the same payload conditioning. The authors should provide an ablation comparing GapONet to a simpler, non-operator network that takes the same inputs (sim context and payload) and outputs the residual action. Without this, it is difficult to ascertain if the performance gain is due to the operator learning formulation or simply the nonlinear, payload-conditioned residual structure.

2. The paper focuses heavily on payload variation. While this is a critical source of non-linearity, the true test of an operator network is its ability to generalize across different functional inputs. The current evaluation only tests generalization across a continuous parameter (payload mass).

3. he paper mentions the curation of the TWINS dataset, which is a major contribution. However, the paper does not explicitly state whether the dataset and the trained GapONet models will be made publicly available. Given the scale and complexity of the data collection, the lack of public release significantly hinders the reproducibility of the results and limits the impact of the dataset contribution.

4. The paper's primary focus is on a model-based correction approach. A key alternative for sim-to-real is robust policy learning via Domain Randomization (DR). The paper should include a more direct and quantitative comparison to a strong DR baseline, where the policy is trained with randomization over the payload range, to demonstrate the superiority of the GapONet correction approach in terms of sample efficiency or final performance.

**Questions:**

1. Ablation on Operator vs. MLP: Could the authors provide an ablation study comparing the proposed GapONet architecture against a standard, high-capacity MLP that is also conditioned on the payload and the simulation context? This is crucial to isolate the performance benefit derived specifically from the operator learning framework.

2. Generalization to New Tasks: The current experiments focus on generalization across payload mass for a fixed set of motions. Can the authors comment on or provide results for the generalization of a trained GapONet to a completely new motion or task that was not part of the TWINS training set?


3. Computational Overhead: What is the inference time overhead introduced by GapONet on the real hardware? Given that the correction is applied at the control frequency, the latency is critical. Please provide a quantitative measure of the inference time compared to the control loop frequency.

4. Role of the Branch-Trunk Decomposition: The paper mentions the branch-trunk decomposition. Could the authors elaborate on the specific functional form learned by the trunk network? Is the trunk network primarily learning the payload-independent dynamics, and the branch network the payload-dependent non-linearities, as hypothesized? A visualization or analysis of the learned functions would be highly informative.

---

> ### Author Response · Authors · 2025-11-21
> **Response to the Reviewer (1 / 2)**
>
> We appreciate your recognition of our work as "highly original" and "compelling," as well as the valuable questions and suggestions you raised—particularly regarding additional experiments and feature analysis, which we believe have further strengthened the soundness of our study. We address your concerns as follows:
>
> ### **[W1 & Q1 Ablation on Operator vs. MLP]**
> Following your suggestion, we conducted an ablation comparing GapONet with a standard high-capacity MLP conditioned on both payload and simulation context, and we further examined the effects of history information and model capacity.
> This resulted in nine additional baselines across four payloads: *MLP-Pointwise-Small/Medium/Large, MLP-History-Small/Medium/Large, and MLP-Sensor-Small/Medium/Large.*
> Both the tables and error curves clearly show the advantage of the operator-based approach, the table in general response is three large-sized models. Please refer to **Supplementary Appendix A.7.3** for more results and detailed analysis.
>
>
> ### **[W2 & Q2 Generalization to Zero-shot Motion]**
> As noted in the section title “Zero-shot Motion Tracking” and in Line 386—*all evaluation trajectories are unseen target joint-position sequences* that never appear in the training data. We have added a clearer description of zero-shot motion in **Lines 403–405** and included a t-SNE visualization of the train/test distributions (position, velocity, and torque) in **Supplementary Appendix A.3.3**. The visualization shows that the distributions barely overlap, further demonstrating that GapONet can generalize across both payloads and motions.
>
> Moreover, as stated in Line 474 and in the captions of Figure 1 and Table 2, all evaluations are performed on a brand-new robot that was never used during data collection. This confirms that our method **generalizes across hardware instances** with small manufacturing differences.
>
> ### **[W3 Open-Source Release]**
> **We will release all data and models.** All resources will be made publicly available within one month, pending the required internal approvals. The codebase and dataset have already been fully prepared. We believe that providing a standardized, large-scale real–robot dataset and a reproducible data-collection pipeline will benefit the community and support future research. Additional details have been included in **Supplementary Appendix A.2**.

---

> > ### Author Response · Authors · 2025-11-21
> > **Response to the Reviewer (2 / 2)**
> >
> > ### **[W4 Comparison to Domain Randomization]**
> > We fully acknowledge that domain randomization (DR) is a widely adopted strategy for sim-to-real transfer. To ensure a fair comparison, we implemented DR baselines for both experiments and included the results in **Table 1, Appendix A.7.4, and videos in the supplementary materials**. These results clearly show that DR struggles to perform the real motions purely through simulator-domain expansion, and the experiments become infeasible as the payload increases.
> >
> > DR and our method address sim-to-real in fundamentally different ways. GapONet explicitly learns the dynamical discrepancy between simulation and reality and outputs a delta action that corrects this mismatch, whereas DR broadens the simulator’s parameter distribution for robustness and does not directly model or correct the structural sim-to-real gap. Consequently, DR cannot offer the targeted correction that GapONet provides, especially in zero-shot and high-payload regimes. Additional analysis is provided in **Appendix A.7.4**.
> >
> >
> > ### **[Q3 Computational Overhead]**
> > This is an important consideration for real-world deployment. The control loop typically requires a frequency of around 50 Hz[1]. We benchmarked the per-loop inference time of GapONet, the MLP baseline, and the Transformer baseline; as shown in Appendix A.7.5 and the table below, all methods operate far above the 50 Hz requirement and therefore meet real-time execution constraints. We also provide real-robot **inference code (model loading + ROS command streaming) in the supplementary materials**, and the remaining components will be released within one month pending approval.
> >
> > | Method   | MLP       | Transformer | GapONet   |
> > |----------|-----------|-------------|-----------|
> > | Time (s) | 0.0001600 | 0.0001181   | 0.0003764 |
> >
> >
> > [1] He T, Luo Z, He X, et al. Omnih2o: Universal and dexterous human-to-humanoid whole-body teleoperation and learning[J]. arXiv preprint arXiv:2406.08858, 2024.
> >
> >
> > ### **[Q4 Role of the Branch-Trunk Decomposition]**
> > We study how the Branch–Trunk decomposition adapts to different inputs by randomly sampling initial states and action sequences, executing them, and comparing Branch Net and Trunk Net outputs across payload masses under identical conditions. We also track how the Trunk Net output changes across the action sequence under a fixed payload. The results show that the trunk network captures payload-independent dynamics while the branch network models payload-dependent nonlinearities. Supporting experiments and visualizations are included in **Appendix A.7.7.**
> >
> > We hope our response has thoroughly addressed your concerns. We are open to further discussion regarding any remaining questions or concerns. If our response and additional results have addressed your concerns, we would greatly appreciate your consideration of a higher score. Your suggestions are instrumental in improving the quality of our paper, and we sincerely thank you for providing your valuable feedback.

---

### Author Response · Authors · 2025-11-21
**General Response (1 / 2)**

We thank reviewer R4BQ, unGq, and DFMY for their valuable feedback, especially highlighting the originality and novelty of framing sim-to-real correction as a nonlinear operator-learning problem (R4BQ, DFMY), the strong motivation and solid theoretical grounding of our formulation (unGq), and the value of the TWINS dataset(R4BQ, DFMY). We follow comments and suggestions from all reviewers and revise our manuscript (colored in blue). We summarize the revisions and supplementary experimental results as follows:

1. **Ablation on Operator vs. MLP.**
We added nine ablation experiments across different inputs, architectures, and parameter counts, showing that the operator’s learning ability comes from the operator itself rather than the residual structure. The results also confirm its better zero-shot generalization over pointwise MLPs, particularly under larger payloads. Detailed results and analysis are provided in **Appendix A.7.3**, with representative tables shown below.

| Method               | LGR (%) ↓ (0kg)         | IQR ↓ (0kg)           | Range ↓ (0kg)         | LGR (%) ↓ (1kg)         | IQR ↓ (1kg)            | Range ↓ (1kg)          |
|----------------------|--------------------------|-------------------------|-------------------------|---------------------------|--------------------------|--------------------------|
| MLP-Pointwise-Large  | 0.08±0.05               | 0.097±0.008            | 0.653±0.090            | 0.76±0.88               | 0.206±0.012             | 0.665±0.059             |
| MLP-History-Large    | 0.11±0.05               | 0.111±0.007            | 0.675±0.098            | 1.17±1.26               | 0.197±0.010             | 0.674±0.058             |
| MLP-Sensor-Large     | 0.09±0.05               | 0.093±0.009            | 0.651±0.076            | 0.87±1.03               | 0.128±0.007             | 0.572±0.066             |
| **GapONet (Ours)**    | **0.09±0.03**           | **0.093±0.016**        | **0.449±0.117**        | **0.22±0.11**           | **0.115±0.013**         | **0.537±0.148**         |


| Method               | LGR (%) ↓ (2kg)         | IQR ↓ (2kg)            | Range ↓ (2kg)          | LGR (%) ↓ (3kg)         | IQR ↓ (3kg)            | Range ↓ (3kg)          |
|----------------------|---------------------------|-------------------------|--------------------------|---------------------------|--------------------------|--------------------------|
| MLP-Pointwise-Large  | 2.19±1.23                | 0.200±0.011            | 0.780±0.075             | 10.76±1.53              | 0.355±0.011             | 0.976±0.096             |
| MLP-History-Large    | 2.43±1.16                | 0.199±0.009            | 0.792±0.077             | 10.74±1.57              | 0.358±0.011             | 0.999±0.104             |
| MLP-Sensor-Large     | 2.66±1.27                | 0.208±0.009            | 0.607±0.067             | 12.05±1.25              | 0.458±0.010             | 0.995±0.114             |
| **GapONet  (Ours)**    | **0.39±0.10**           | **0.161±0.004**        | **0.578±0.112**         | **0.84±0.23**           | **0.317±0.005**         | **0.498±0.157**         |

2. **Experiments of Three New Strong Baselines.**
We additionally incorporated a domain-randomization baseline for sim-to-real transfer and two nonlinear system-identification baselines to further validate the effectiveness of operator learning. The corresponding results and analyses are provided in **Table 1, Appendix A.7.4, Appendix A.8.3, and the supplementary videos.**

3. **Analysis of Train/Test Dataset.**
We detailed the analysis for both the train and test sets, along with t-SNE visualizations of position, velocity, and torque in **Appendix A.3.3**, which directly validates the correctness of our zero-shot setup and further demonstrates that GapONet generalizes across payloads, motions, and even new hardware instances (robot).

4. **Mathematical symbols and details.**
To improve readability and in response to Reviewer DFMY’s suggestion regarding reorganization, we added more detailed explanations of the **symbols in Section 4**, and expanded **Appendix A.3.3 (train/test data split)**, **Section 5 & Appendix A.7.6 (experiment setup)**, and **Appendix A.8.3 (baseline execution details)**. We also included the **algorithm for our method in Appendix A.6** and provided a complete list of **symbol definitions in Appendix A.6.2.**

---

> ### Author Response · Authors · 2025-11-21
> **General Response (2 / 2)**
>
> 5. **Detailed Description of GapONet Training.**
> - We analyzed the roles of the branch and trunk networks during training, showing that the trunk network captures payload-independent dynamics while the branch network models payload-dependent nonlinearities. Supporting experiments and visualizations are included in **Appendix A.7.7.**
> - We clarified the design of the LGR metric and explained how LGR, IQR, and Range each assess different aspects of the sim–real discrepancy (threshold-based proportion vs. distribution spread). Experiments in Appendix A.7.3 help to demonstrate the usefulness of all three metrics.
> - We elaborated on why reinforcement learning is necessary: the optimization depends on non-differentiable simulator dynamics and requires GPU-parallel execution for feasible training. This makes RL the only practical and efficient choice (see **Appendix A.6.3**).
> - We also clarified the distinction between sim–sim and sim–real gaps in Figure 3, emphasizing that the substantially larger sim–real gap highlights the importance of learning this discrepancy.
>
> 6. **Motion Tracking with Object Interaction.**
> To further demonstrate our payload generalization capability, we additionally performed zero-shot interactions with a box and a chair. The corresponding videos are included in the **supplementary materials**.
>
> 7. **Open-Source Release.**
> **We will release all data and models.** We believe that providing a standardized, large-scale real–robot dataset and a reproducible data-collection pipeline will benefit the community and support future research.

---

### Author Response · Authors · 2025-12-02
**Clarifications on Paper #18528**

Dear Area Chair,

We sincerely appreciate the efforts the Area Chair and Senior Area Chair are investing to ensure a fair evaluation in this unforeseen circumstance. We summarize our submission to hopefully alleviate the AC and SAC's workload.

Our paper presents:
- TWINS, the first paired **sim–real dataset on full-size humanoids across three simulators**, designed to systematically analyze payload-induced sim-to-real discrepancies (11,298\*4 sequences; 30*4+ hours).
- GapONet, a payload-conditioned nonlinear operator that outputs real-time residual actions. It maintains **sim–real alignment under increasing payloads** with only 0.09% of frames exceeding the large-gap threshold.

To address the reviewers’ concerns, we focused on targeted revisions: 9 ablations contrasting operator learning with pointwise models, 3 additional baselines, expanded train/test analyses with t-SNE visualizations, and clearer descriptions of baseline implementations and notation. The experimental settings requested by the reviewers have also been clarified in the manuscript (highlighted in blue).
The **General Response** below addresses each comment point-by-point with references to the revised sections.

With all reviewers reporting a confidence of 3, we understand that some technical aspects may not have been fully conveyed initially and additional clarification is important. We have provided it with targeted revisions, and we appreciate an evaluation based on the clarified technical contributions. Thank you for your time and consideration.

Yours sincerely,

Authors of paper #18528

---

### Meta-Review · Area_Chair_Qzwf · 2026-01-11

**Summary:**

This paper targets the humanoid sim-to-real gap that becomes especially severe under end-effector payload changes during loco-manipulation. It proposes GapONet, a payload-conditioned nonlinear operator that maps simulation “context/actuation functions” to residual actions applied on hardware for real-time compensation. Across the reviews, the work is viewed as important and promising, especially due to the dataset/pipeline and the framing of payload as a structured conditioning variable rather than random disturbance. However, the current submission does not yet meet the ICLR bar for clarity and validation of its central claim: that the operator-learning formulation provides distinct benefits beyond “a strong nonlinear residual network conditioned on payload.” Additionally, the paper leaves key practical questions under-answered (latency/inference overhead at control rates, the necessity of RL vs supervised regression, baseline training protocol details, and dataset/model release). After carefully reading the paper, review and author responses, the AC agrees with the majority of the reviewers on rejecting the paper.

**Reviewer Concerns:**

see Summary

**Reviewer Scores:**

see Summary

---

### Decision · Program_Chairs · 2026-01-26

Reject